# Continuous or discrete magnitudes? A comparative study between cats, dogs and humans

Mireia Solé Pi[1], Luz A. Espino[1], Péter Szenczi[2,3], Marcos Rosetti[1,3]*, Oxána Bánszegi[1]*

**1** Instituto de Investigaciones Biomédicas, Universidad Nacional Autónoma de México, Mexico City, Mexico, **2** Consejo Nacional de Humanidades, Ciencia y Tecnología, Mexico City, Mexico, **3** Instituto Nacional de Psiquiatría Ramón de la Fuente Muñiz, Unidad Psicopatología y Desarrollo, Mexico City, Mexico

☯ These authors contributed equally to this work.
\* oxana.banszegi@gmail.com (OB); mrosetti@gmail.com (MR)

## Abstract

A long-standing question in the study of quantity discrimination is what stimulus properties are controlling choice. While some species have been found to do it based on the total amount of stimuli and without using numerical information, others prefer numeric rather than any continuous magnitude. Here, we tested cats, dogs, and humans using a simple two-way spontaneous choice paradigm (involving food for the first two, images for the latter) to see whether numerosity or total surface area has a greater influence on their decision. We found that cats showed preference for the larger amount of food when the ratio between the stimuli was 0.5, but not when it was 0.67; dogs did not differentiate between stimuli presenting the two options (smaller vs. larger amount of food) regardless of the ratio between them, but humans did so almost perfectly. When faced with two stimuli of the same area but different shapes, dogs and humans exhibited a preference for certain shapes, particularly the circle, while cats' choices seemed to be at chance level. Furthermore, cats' and dogs' reaction times were equal across conditions, while humans were quicker when choosing between stimuli in trials where the shape was the same, but the surface area was different, and even more so when asked to choose between two differently sized circle shapes. Results suggest that there is no universal rule regarding how to process quantity, but rather that quantity estimation seems to be tied to the ecological context of each species. Future work should focus on testing quantity estimation in different contexts and different sources of motivation.

**Data availability statement:** Data available in repository: https://dx.doi.org/10.6084/m9.figshare.25632501 and https://dx.doi.org/10.6084/m9.figshare.28489685.

**Funding:** This work was supported by the Programa de Apoyo a Proyectos de Investigación e Innovación Tecnológica of the Dirección General de Asuntos del Personal Académico, Universidad Nacional Autónoma de México (https://dgapa.unam.mx/) [PAPIIT, DGAPA, grant number IA204222] to O.B. and by a Cátedra grant to P.S. from the Consejo Nacional de Humanidades, Ciencia y Tecnología, México (https://conahcyt.mx/) [CONAHCyT, grant number 691] to P.Z. Funders did not play any role in the study design, data collection and analysis, decision to publish, or preparation of the manuscript.

**Competing interests:** The authors have declared that no competing interests exist.

## Introduction

A long-standing debate in cognitive research has focused on quantity discrimination. The 'number sense' theory states that animals have an innate, basic ability to perceive, understand and discriminate using numerosity (processing of discrete magnitudes), while the 'sense of magnitude' theory states that the difference in continuous magnitudes (non-discrete, infinitely divisible properties of stimuli such as surface area, mass, or perimeter) is the driving force behind quantity discrimination [1–3]. Continuous magnitudes also covary with numerosity; therefore, it is impossible to generate sets of stimuli that have differences only in numerosity and no other properties [2,4].

It has been shown that even when controlling for certain magnitudes (i.e., size) in numerical comparison tasks, others (i.e., cumulative area) are likely to influence choice. EEG studies conducted on human adults showed that relevant brain areas are more active when reacting to continuous magnitudes than to numerosity, even when the participant is asked to focus on the numerosity of the stimuli [5]; additionally, number change is processed slower than shape change in the cognitive processing stream [6]. This suggests that, while we can use both, the default way in which we estimate quantities is by using non-numerical variables.

Studies investigating non-symbolic number processes on non-human animals (animals from now on) have drastically increased in the past decades [review in 7–12]. It is well known that animals can differentiate between quantities depending on continuous magnitudes such as size or surface area, but also between sets of different numbers of items. This ability can be advantageous for any animal in diverse life situations and ecological contexts, such as optimizing food intake [13–15], reducing predation risk by joining groups with more individuals [16–18], or increasing mating chances by joining groups with a more advantageous sex ratio [19]; thus, it should be favored by natural selection. However, when the continuous magnitudes and numerosity of the items coincide, the strategy the animals use to make their decision becomes less clear.

This is in line with the 'last resort' hypothesis, which proposes that processing numerical information involves a larger cognitive load than estimating continuous magnitudes. Given that using a discrete property requires more effort to process compared to other properties of a resource, the numerosity of the stimuli should be the last feature to be used during discrimination [20,21]. Nonetheless, a study employing a numerosity comparison task solved by human participants and deep neural networks reported that both humans' and the networks' choices were mainly influenced by the numerosity of the stimuli, despite the presence of other non-numerical cues [22]. These studies found that continuous visual features, particularly total item perimeter, modulated this characterization of numerosity; even so, the authors' results seem to support the theory that a fast and automatic 'number sense' is the primary tool for quantity discrimination in adult humans.

The 'last resort' hypothesis is not unanimously supported by experiments which tested animals' preferences through choice between different numbers of stimuli but

controlled for volume or surface area, or the other way around. While some species have been found to discriminate between quantities on the basis of the total amount of stimuli and without using numerical information, such as cats [23], dogs [24,25], and crows [26], others have been found to exhibit a preference to choose based on number rather than on continuous magnitudes of the stimuli, for example rhesus monkeys [27,28], horses [29], and orb-weaver spiders [30]. Chimpanzees have also been found to base their judgment on the total amount of stimuli rather than on number or contour length; however, they also preferred the set containing the largest individual food item despite it having a lower total amount of stimuli [31].

Few studies directly compare the use of numerosity with the use of continuous magnitudes of stimuli when discriminating between quantities; the ones that do tend to be focused on one single species, entailing methodological differences (e.g., training or spontaneous choice, artificial stimuli or food) that make comparison between species a difficult task. Research from the last decades suggests that the domestic cat (*Felis silvestris catus*) and dog (*Canis lupus familiaris*) are interesting model species for comparative non-invasive neuroscience. Several aspects of cat [32] and dog cognition [33,34] have been intensively investigated. Studies focusing on cat's and dog's quantitative abilities found that both species rely on the difference between the numerosity or surface area of the stimuli in their decision, meaning that as the ratio between two stimuli increases, discrimination becomes harder. While cats can spontaneously discriminate up to a ratio of 0.5 [13], dogs can do it up to a 0.67 ratio [24,35]. With training, cats are able to discriminate up to a 0.67 ratio [23] and dogs up to 0.80 [25]. In addition, adult cats can differentiate between the number and size of live prey [14].

Cats and dogs appear to use non-numerical continuous magnitude as the basis for discrimination when stimulus sizes differ. Pisa and Agrillo [23] trained adult cats to discriminate between sets of two and three same-sized dots on a screen. With this training, cats reliably learned to choose the group with the greater numerosity. However, when the surface area was controlled, cats' response to the two stimuli dropped to the level of chance, suggesting their previous preference was based on the surface area and not on number, or that both cues were being used simultaneously and changing one of them hindered proper discrimination. Additionally, cats reliably chose the larger amount of food when the ratio between the two stimuli was greater than 0.5, both when the difference between stimuli was in their numerosity and in their surface area.

Similarly, Miletto Petrazzini and Wynne [24] tested adult dogs' spontaneous discrimination between different amounts of food. They found that dogs reliably chose the set with more food items, or the set with the larger amount of total food even if it was presented as a smaller number of items. Nonetheless, when the total amount of food between sets was equal but the number of food items differed, or when the number of items was equal, but they varied in individual and total surface area, the dogs' choice did not differ from chance level.

Cat and dog behavior is often studied separately, despite these organisms having similar relationships to humans and living in similar environments. Therefore, our aim was to test cats and dogs in a simple food-based two-way spontaneous choice paradigm to determine whether numerosity or total surface area of the stimuli had a greater influence on the animals' choice. This would allow us to directly compare the quantity discrimination skills of the two most common pet species, using a new protocol that controls for shape as well as surface area.

We propose two control conditions to assess the subjects' capacity to differentiate between sets of food with different surface areas presented simultaneously, and an experimental condition in which the sets of food had the same surface area but were presented in different shapes; these shapes varied in their numerosity and individual area. Additionally, we conducted a similar experiment on human participants to provide a baseline with the findings on cats and dogs, as the main theories for quantity representation are based on human decision-making abilities [36].

## Methods

### Subjects

**Cats and dogs.** Pet owners interested in participating in the study were recruited through social media and a network of students and acquaintances. Inclusion criteria when selecting the animals were: a) friendly with strangers, b) free from

evident vision problems, c) at least 1 year old and, d) in the case of dogs, at least 10 kg of weight. This last constraint was included to try to keep a consistent ratio between the size of the dog and the food portions, as well as to maintain a similar angle from which they observed the stimuli. We also avoided smaller dogs as they may quickly get satiated. Additionally, we collected consent from the owner to deprive their pet of food for 4 h prior to each session, as well as to allow the experimenters to visit and test their pet on at least two separate occasions.

We visited a total of 36 domestic cats and 38 domestic dogs at their homes in Mexico City, Estado de Mexico and Morelos, all in central Mexico. From this, a subset of eight cats and three dogs had to be discarded because they were too nervous/anxious (e.g., hiding from the experimenter, freezing in the same spot, not leaving their owner's side), and/or showed lack of motivation to go after the food stimuli on three consecutive trials. An additional two cats and six dogs were not included in the data analysis because the animals showed a lateral bias. That is, they chose the same side regardless of the stimuli presented in at least five consecutive trials on three consecutive testing sessions or in at least 11 out of 14 trials in one session. The final dataset is shown on Table 1. Details of each of the individual animals tested can be found in an online repository: http://doi.org/10.6084/m9.figshare.28489685

**Humans.** We recruited young adults studying at the Universidad Nacional Autónoma de México (UNAM), in Mexico City through flyers, social media and word of mouth. Inclusion criteria for participation were a) have normal or corrected vision, b) be native Spanish speakers, c) have between 18 and 26 years of age, and d) not be under the influence of any psychoactive substance. The experimenter confirmed these points before the session. If they agreed to participate, they signed a consent form. A total of 40 participants completed the tests; however, the data from two of them had to be excluded from later analysis, since they verbally expressed to have misunderstood the instructions. The final dataset consisted of 38 participants (age mean ± SD = 21.65 ± 2.26 years, range 18–26 years; 14 males and 24 females). Recruitment took place between March 27 and August 25, 2023.

## Procedure

**Cats and dogs.** Fig 1 displays the shapes employed and Fig 2 contains the setup used. Each animal completed 14 spontaneous two-way food choice trials which comprised three different conditions, two control conditions (designed to see if the animal could differentiate between two different amounts of food presented simultaneously) and one experimental condition (the same amount of food presented in different shapes).

Cats show marked individual differences in food preference [13,37–39]. Thus, to ensure that they were motivated for the duration of the trials, prior to the start of the first session, we tested each individual with three choices (i.e., their usual canned food, ham, or canned tuna) to find their preferred food item. The first food eaten was the stimulus used in all trials with that individual. In our previous experiments, we have also found it beneficial to use the animal's preferred food to keep it motivated [13,37,40,41]. Of the total 26 cats: nine were tested with canned cat food, five with canned tuna, and 12 with ham. For dogs, higher quality foods have been shown to induce greater motivation than kibble [42]. Prior to the first

Table 1. Details of the samples of cats and dogs tested.

| Spp. | Age (years, mean ± SD [range] | Weight (kg, mean ± SD [range]) | Sex | | | | Breed (n) | Lifestyle (n) |
|---|---|---|---|---|---|---|---|---|
| | | | Male | | Female | | | |
| | | | Neu-tered (n) | Intact (n) | Neu-tered (n) | Intact (n) | | |
| Cats (n = 26) | 6.42 ± 3.74 [1-14] | – | 12 | 4 | 9 | 1 | All mixed | 14 exclusively indoor, 12 indoor/outdoor |
| Dogs (n = 28) | 4.06 ± 3.1 [1-13] | 23.4 ± 8.4 [10-40] | 9 | 4 | 11 | 4 | 16 mixed, two bulldogs, two Belgian shepherds, two German shepherds | 12 trained to follow verbal commands |

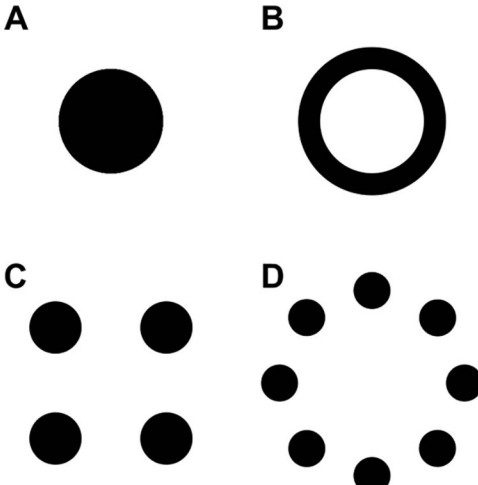

**Fig 1. Stimuli used in the task.** Stimuli (either as food or as geometric shapes) were presented in four different shapes. Spatial arrangements used in the tasks: a) one large, filled circle (circle); b) a hollowed-out circle (donut); c) four same-sized small circles (dots); and d) eight same-sized small circles arranged on the corners of a regular octagon (crown).

session, dogs were offered ham; all subjects accepted it and begged for more, so we proceeded to perform the trials with this type of food.

For comparative purposes, the methods were similar to those previously used with cats [13,41] and dogs [24,43] to assess their quantity discrimination abilities and visual illusion perception with some slight modifications (see below). Food stimuli were presented in four different arrangements which were all flat. These four arrangements had the same surface area, but contained different number of elements (as shown in Fig 1): a) one large full circle (circle), b) a hollowed-out circle (donut), c) four same-sized small circle (dots) or d) eight same-sized small circles, each placed at the corners of a regular octagon (crown). We added the donut shape to test a round stimulus with the same properties and numerosity as the circle, but a larger visual complexity as it contains a subtraction of area in the middle.

We chose three different diameters (d) which corresponded to three different surface areas (a) to display the food for the cats (large: d = 4 cm, a = 12.56 cm$^2$; medium: d = 3.27 cm, a = 8.42 cm$^2$; and small; d = 2.82 cm, a = 6.2 cm$^2$) and for the dogs (large: d = 6.32 cm, a = 31.4 cm$^2$; medium: d = 5.18 cm, a = 21.04 cm$^2$; and small: d = 4.47 cm, a = 15.7 cm$^2$).

In the control conditions, we offered the animals the same shape of food, but with different surface areas, to see whether they would spontaneously choose the larger portion of food. In Control A (four trials) the ratio between the surface areas of the two food portions (large vs. small) was 0.5, thus the surface areas were the following: 12.56 cm$^2$ vs. 6.28 cm$^2$ for the cats, and 31.4 cm$^2$ and 15.7 cm$^2$ for the dogs. In Control B (four trials) the ratio between the surface areas of the two food portions (large vs. medium) was 0.67, thus the surface areas were the following: 12.56 cm$^2$ vs. 8.42 cm$^2$ for the cats, and 31.4 cm$^2$ vs 21.04 cm$^2$ for the dogs. In the experimental trials, we presented food portions of equal size (large) in pairs but arranged in different shapes (six trials): circle vs. donut, circle vs. dots, circle vs. crown, donut vs. crown, donut vs. dots, and crown vs. dots.

Food stimuli were presented on two flat black square plastic plates (size 12.5 cm x 12.5 cm for the cats and 20 cm x 20 cm for the dogs), which were placed on the floor on a matte grey plastic sheet (47 cm x 68 cm for the cats and 100 cm x 60 cm for the dogs), 15 cm apart for the cats and 30 cm apart for the dogs (distance between the inner edges of the black sheets). A chair was used to give the cats a starting point with a non-distorted overview of the stimuli (chair height: 35–42 cm, placed 25 cm from the edge of the grey sheet), something dogs did not need due to their height.

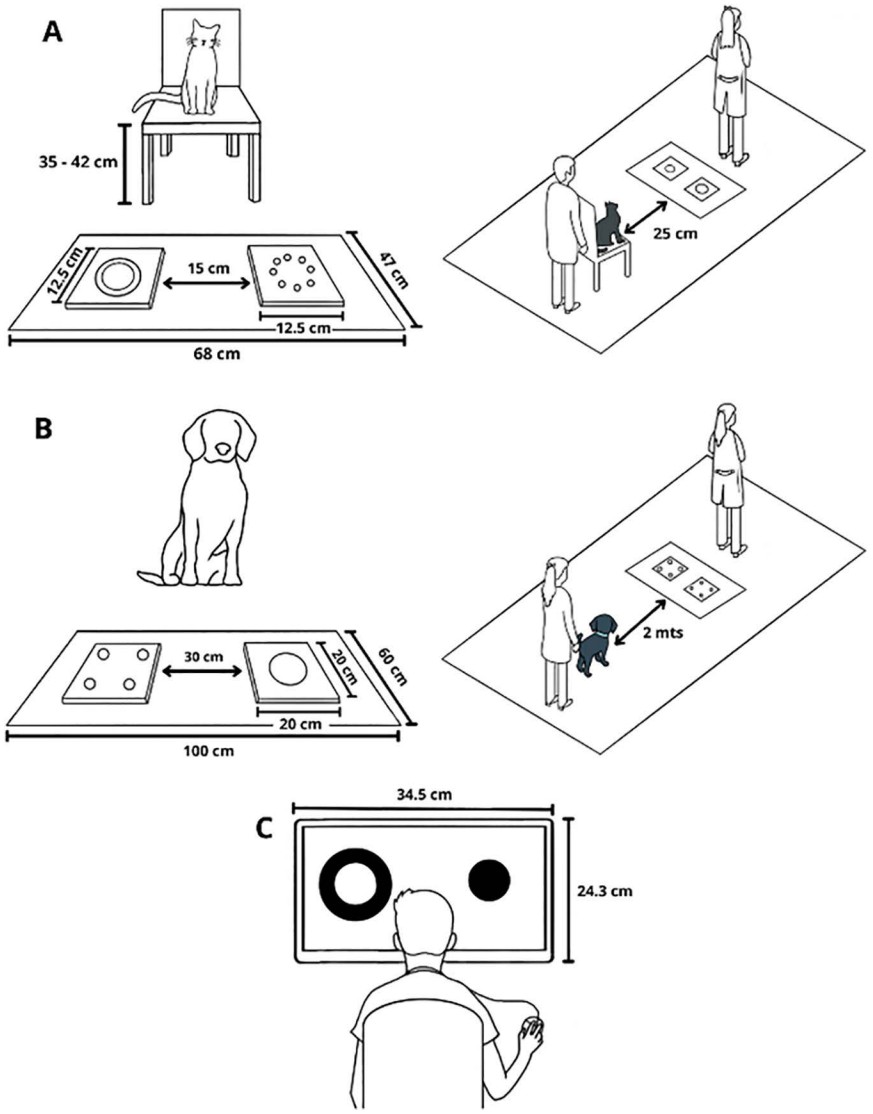

**Fig 2. Experimental setup.** Schematic overview of the two-way spontaneous choice task for A) cats, B) dogs, and C) humans. For cats and dogs, the left panel shows the presentation of stimuli. The right panel shows the position of the animal subject, owner, and experimenter relative to the stimuli: subjects were placed at a marked position by their owners facing the sheets and released after 5 s of viewing the stimuli. Human participants completed the task on a computer screen with a mouse.

Animals were tested individually in their homes, preferably in a closed room which they were familiar with. Tests were conducted between 09:00 and 19:00 h according to the owners' availability. Out of sight of the owner and the animal, the food stimuli were prepared on the black plates and placed on the grey sheet. Then, the experimenter turned on the camera (GoPro 4 Session camera, GoPro Inc, California) and walked away but stayed aligned with the middle of the plastic sheet.

The experimenter then asked the owner to come in with the animal, place it on the starting point, and gently hold it with their hands in the same spot from behind for 5 s to give it the chance to view both plates before releasing it to make its choice. To avoid cueing, both the owner and the experimenter looked upwards so their gaze would not provide any signals to the animal. For the cats, the starting point was a chair facing the stimuli 25 cm away from the grey sheet. For the dogs, it was a

point marked point on the floor 2 m away from the grey sheet. We defined "making a choice" as the animal approaching one of the plates and manipulating the food (i.e., licking, eating, or pawing). As soon as the animal chose a plate, the experimenter removed the other plate and allowed the animal to eat briefly from the chosen stimulus to keep it from getting satiated.

Some of the cats lost motivation (i.e., stopped making choices), thus were given a 10-minute break. If the cat continued to be uninterested, the session was suspended and resumed on additional days until the cat had completed all the trials. None of the dogs lost motivation or stopped making choices.

In order to exclude any side preferences, we counterbalanced the left-right presentation of the small/large portions and the different shapes of food across trials. We also rotated the location of the stimuli and starting point in the room to avoid the use of environmental cues (e.g., a specific window or a door). The order of the trials (controls and experimental) between sessions and participants was randomized.

**Humans.** After participants signed the consent form, the experimenter asked for the participant's initials, age, sex, and if their vision has been corrected or not. Subsequently, they were taken to a quiet and well-lit room and instructed to sit at a desk with a computer and a mouse. The experimenter asked the participant to carefully read and follow the instructions shown on the screen. The experimenter stayed in the room during the task, sitting opposite from the participant and facing away so as not to influence or distract them. The computer alerted the experimenter when the task was done. The task took about five minutes to complete.

We used a series of two-way choice tasks with pictures as stimuli which were created in the PsychoPy software [44] and displayed on a 14-inch LCD screen of a Dell Vostro 14 3000 computer with a mouse. Instructions presented on the screen asked the participant to click on the image with the larger black area as fast as they could (full verbal and written instructions can be found in S1 File). Each person, just like the animals, completed trials of three different conditions: two controls (to see whether the participant understood the task) and one experimental condition. Since trials were simple and brief, we were able to present two versions of each trial by switching the sides (left/right) of the stimuli, giving participants a total of 28 trials to complete: 8 for Control A, 8 for Control B and 12 for the experimental condition. The presentation order of all the trials was randomized for each participant.

In the control conditions, the same shapes, but with different surface areas were presented. In Control A, the surface areas were 29.22 cm$^2$ vs. 14.61 cm$^2$, thus, the ratio between the two stimuli was equal to 0.5; in Control B the surface areas were 29.22 cm$^2$ vs. 19.58 cm$^2$, thus the ratio between the two stimuli was equal to 0.67.. For the trials of the experimental condition, we used stimuli with equal surface areas but presented as different shapes (12 trials) in the same was as we did for the animals.

The participants' choice and latency were automatically recorded for each trial. If participants took more than 10 s to click on either option (i.e., the maximum choice latency allowed), the trial was cancelled and the next trial started immediately, recording the data for the skipped trial as missing.

## Ethics statement

The procedures described in the current manuscript were approved by the *Comisión Institucional para el Cuidado y Uso de Animales de Laboratorio* with ID 10384 and the *Comité de Ética en Investigación con Seres Humanos*, both of which belong to the Instituto de Investigaciones Biomédicas of the Universidad Nacional Autónoma de México and adhere to strict ethical guidelines for animal subjects and human participants, respectively. Written consent from pet owners and human participants was provided before any testing was performed.

## Analysis

All analyses were performed in R (version 4.3.2). For control conditions, we ran two binomial tests for each species, one for Control A and one for Control B, comparing the number of trials where the larger portion had been chosen to the total number of control trials across all subjects of that species, with $\alpha = 0.5$ (chance level). We performed additional binomial

tests to compare the number of choices for the larger option according to the shape of the stimuli (circle, crown, donut or dots), to identify if any shapes resulted in better area discrimination. We also performed two Wilcoxon tests, one to compare performance between Control A and B trials, to determine whether the larger portion was chosen more often in one of the two controls, and one to compare performance between the four stimuli shapes, to determine whether there were differences in choice depending on the shape of the control.

For the experimental condition, we ran six binomial tests for each species, one for each possible combination of stimuli (e.g., circle vs. crown), comparing the number of trials where either shape was chosen, with $\alpha = 0.5$ (chance level). The effects of sex, age, weight, and other possible confounding variables (e.g., order of presentation) were calculated by building binomial logistic Generalized Linear Models (GLMs) for each species (one for combined control trials, one for experimental trials and one for all trials) with the proportion of choices as a response variable, and with these demographic variables as predictors.

Latencies for each species were analyzed by means of three pairwise Wilcoxon tests, one test comparing the latency values according to the type of condition (Control A, Control B, or Experimental) to determine whether subjects were faster or slower depending on the condition, one test comparing the latencies according to the shape of the control (e.g., large circle vs small circle, large crown vs small crown) to determine if any shapes resulted in a faster total area discrimination, and another test comparing the latency values according to the combination of shapes presented in the experimental trials (e.g., circle vs dots, donut vs crown) to determine whether any particular combination of stimuli influenced choice latency.

## Results

### Control (size) conditions

As stated previously, from the original sample of 36 domestic cats and 38 domestic dogs, only data from 26 and 28, respectively, were analyzed. From these subjects, we found that cats chose the larger amount of food 69/104 times (0.66) in Control A and 61/104 times (0.58) in Control B. Binomial tests showed that, on the control conditions, the choice of the larger shape surpassed chance levels in A ($p = 0.001$, 95%CI = [0.56, 0.75]), but not B controls ($p = 0.09$, 95%CI = [0.46, 0.68]). In Control A, the crown was the only shape where the choice for the larger stimuli surpassed chance levels, according to individual binomial tests ($p = 0.02$, 95%CI = [0.52, 0.88]); in Control B, none of the shapes surpassed chance level. There was no significant difference in performance according to shape of the control ($p = 0.05$). A Wilcoxon test showed no difference when comparing the proportion of choices for the larger stimuli between the two ratios ($W = 78.5$; $p = 0.29$; Fig 3A).

Dogs chose the larger amount of food 65/112 times (0.58) in Control A and 50/112 times (0.44) in Control B from; binomial tests showed that the choice for the larger shape did not surpass chance levels in either control (Control A: $p = 0.1$, 95%CI = [0.48, 0.67]; Control B: $p = 0.29$, 95%CI = [0.35, 0.54]). There was no difference in performance according to shape ($p > 0.05$). However, a Wilcoxon test showed that the difference in performance between control conditions was significant ($W = 168$; $p = 0.01$, effect size = 0.24; Fig 3B), with dogs selecting the larger portion slightly more often in Control A as opposed to Control B.

For humans, binomial tests showed that the choice for the larger shape surpassed chance levels for both controls ($p < 0.001$), with them choosing the larger stimuli in 293/301 trials in Control A (0.97; 95%CI = [0.95, 0.99]) and 296/300 trials in Control B (0.98; 95%CI = [0.97, 1.00]) across all subjects. All the shapes surpassed chance levels for both controls, according to binomial tests ($p < 0.001$); since errors were minimal in the trials, there was no difference in performance according to shape ($p > 0.05$). We also did not find a difference when comparing the proportion of choices for the larger stimuli between the two control conditions with a Wilcoxon test ($W = 1.5$; $p = 0.265$; Fig 3C). There were seven trials where choices were not recorded either because the participant did not answer, or because of technical glitches, in which the program failed to record the response.

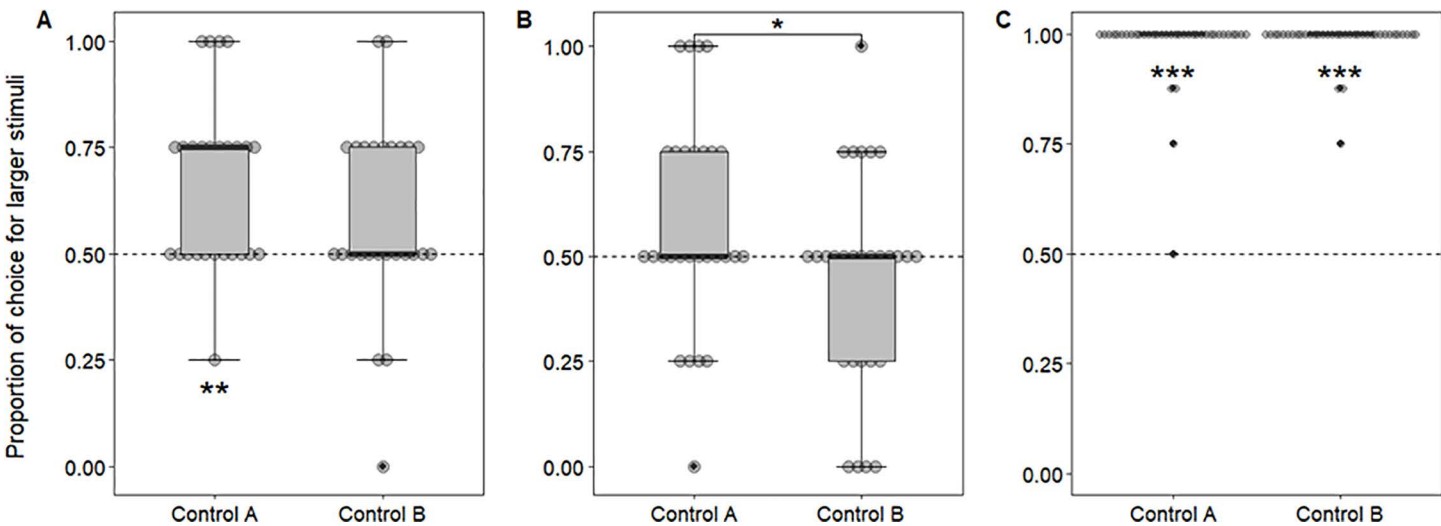

**Fig 3. Proportion of choices.** A) cats', B) dogs', and C) humans' proportion of choices for the larger stimuli in the Control conditions. The thicker line represents the median of the data, while the lower and upper hinges correspond to the first and third quartiles (the 25th and 75th percentiles). The upper and lower whiskers extend 1.5 * IQR (inter-quartile range) from the upper and lower hinges. Data beyond the whiskers are outliers and are represented as black dots. Each subject is represented with a grey circle, the dotted line indicates chance level (0.5), ** corresponds to p < 0.01 and *** corresponds to p < 0.001.

## Experimental (shape) conditions

For cats, none of the shapes were chosen significantly more than the others, according to the six binomial tests (p > 0.05, Fig 4A). For dogs, on the other hand, the binomial tests showed that shape had an effect on choice with subjects showing a significant preference for the circle over the crown (p < 0.01, 95%CI = [0.59, 0.91]; Fig 4B).

For humans, we also observed an effect of shape on choice, with binomial tests revealing a significant preference for the circle over the donut (p < 0.001, 95%CI = [0.72, 0.9]), the crown over the donut (p < 0.001, 95%CI = [0.61, 0.82]), the dots over the donut (p < 0.001, 95%CI = [0.62, 0.83]), and the circle over the dots (p = 0.02, 95%CI = [0.51, 0.73]; Fig 4C).

## Latencies

The mean latency of the cats' and dogs' choices was not influenced by the condition of the trials, according to the pairwise Wilcoxon tests (Table 2). For humans, a pairwise Wilcoxon test revealed that the choice latency in trials in experimental condition was significantly greater than in both controls (Control A vs Experimental: p < 0.001, effect size = 0.45; Control B vs Experimental: p < 0.001, effect size = 0.42; Table 2). According to this test, there were no differences in latency between Control A and Control B conditions (p > 0.05).

When combining the trials from both control conditions, we found that shape of the stimuli in the control did not influence the latency to choose in cats and dogs, according to the pairwise Wilcoxon tests (Table 3). However, for the humans the pairwise Wilcoxon test showed that latency was smaller when comparing two circles with different areas (mean$_{A+B}$ ± SD = 0.97 ± 0.41 s) than for control trials with small vs large versions of all other shapes: crown (p < 0.001, effect size = 0.35); donut (p < 0.01, effect size = 0.27); and dots (p = 0.01, effect size = 0.23). There were no differences in the latency to choose in the pairs of trials in the experimental condition for any of the three species, according to the three Wilcoxon tests.

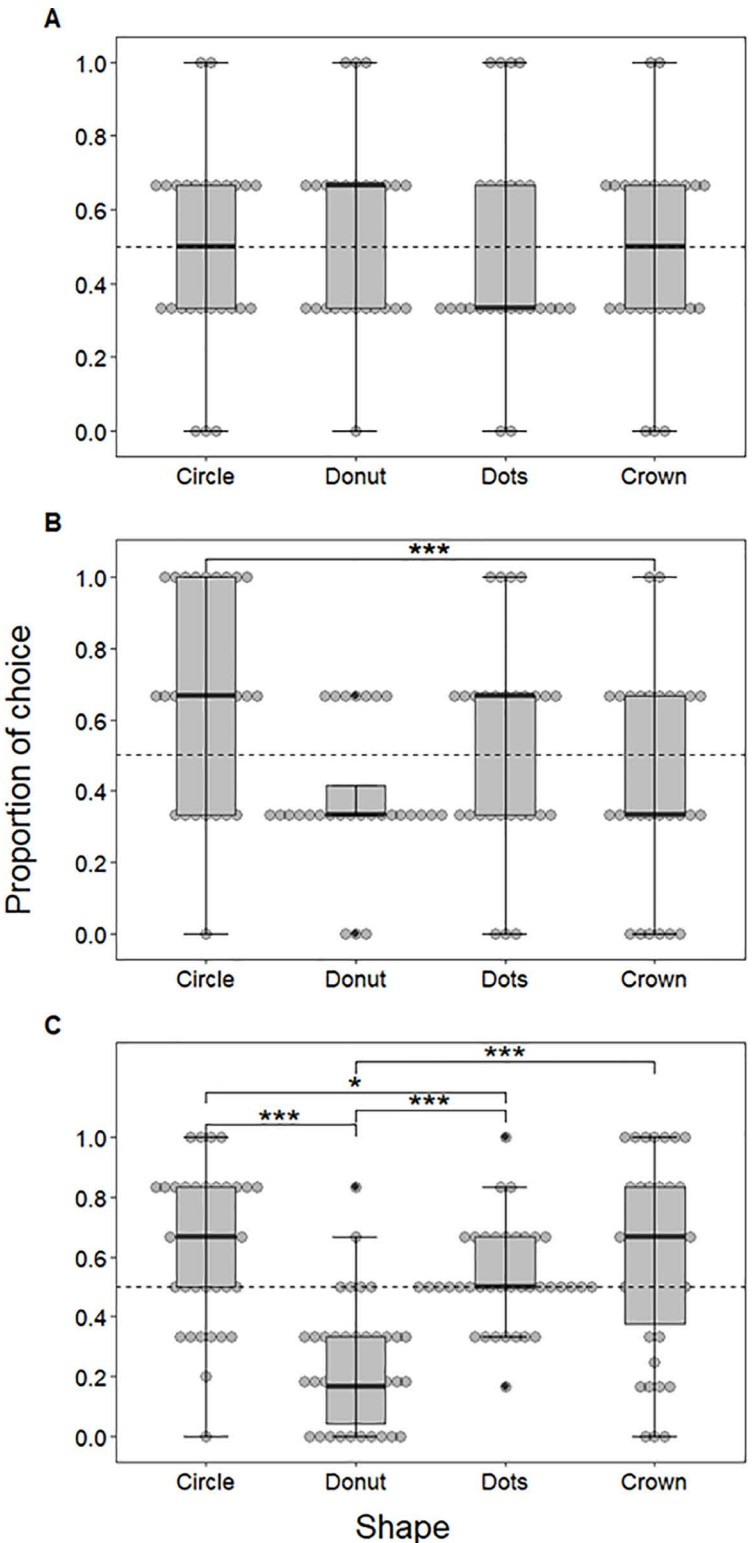

**Fig 4. Proportion of choices per shape.** A) cats', B) dogs', and C) humans' proportion of choices for each shape (circle, donut, dots, and crown) in the experimental condition. The thicker line represents the median of the data, while the lower and upper hinges correspond to the first and third quartiles (the 25th and 75th percentiles). The upper and lower whiskers extend 1.5 * IQR (inter-quartile range) from the upper and lower hinges. Data beyond the

whiskers are outliers and are represented as black dots. Each subject is represented with a grey circle, the dotted line indicates chance level (0.5), * corresponds to p<0.05 and *** corresponds to p<0.001.

**Table 2. The mean latency (s) ± SD of the cats', dogs' and humans' choices in both Control and Experimental conditions.**

|  | Cats | Dogs | Humans |
|---|---|---|---|
| Control A (1:0.5) | 4.71±4.20 | 2.87±1.54 | 1.21±1.1 |
| Control B (1:0.67) | 4.69±2.88 | 2.75±1.76 | 1.21±0.66 |
| Experimental | 4.88±2.86 | 2.68±1.31 | 1.98±1.15 |

**Table 3. The mean latency (s) ± SD of the cats', dogs' and humans' choices in the combined Control conditions (A and B) for each shape.**

|  | Cats | Dogs | Humans |
|---|---|---|---|
| Small circle vs large circle trials | 4.87±3.10 | 2.63±1.44 | 0.97±0.41 |
| Small donut vs large donut trials | 4.54±3.10 | 2.98±1.63 | 1.17±0.46 |
| Small dots vs large dots trials | 4.45±2.35 | 2.85±2.2 | 1.2±0.78 s |
| Small crown vs large crown trials | 4.62±2.14 | 2.77±1.18 | 1.47±1.48 |

## Demographic and other variables

We found that female dogs chose the larger portion more than males on the control conditions (estimate±SE = 1.24±0.61, Z-ratio = 2.23; p = 0.03) after modelling this variable against the responses in both control conditions. Upon dividing the data according to sex, we found that female dogs' choice for the larger portion in Control A (0.66 of trials) surpassed chance levels following a binomial test (p = 0.01, 95%CI = [0.53, 0.79]), but that male dogs' choices did not surpass chance levels for either control. In Control B, female dogs' choice for the larger portion did not surpass chance levels but was still higher than that of male dogs (larger portion chosen 46% of times as opposed to 38% of times for the males). No other effects were found for demographic variables (age and sex for cats and humans; food type and lifestyle for cats; age, weight, and training for the dogs) nor order of presentation for any of the species.

## Discussion

The main aim of the present study was to compare three species – cats, dogs, and humans – in a simple two-way spontaneous choice paradigm to determine whether the numerosity or the continuous magnitudes of stimuli have more influence in their decision. We found that cats showed a significantly more frequent choice for the larger amount of food when the ratio between the stimuli in the control conditions was 0.5, but not when it was 0.67, while dogs did not differentiate between quantities in either of the control conditions. For both control conditions, we found that the human's choice of the larger images was significantly above chance level.

In the experimental condition trials, we found that dogs and humans chose some shapes more frequently, while cats were indifferent between shapes. Furthermore, cats' and dogs' reaction time was equal across conditions, while humans were faster in controls, and even more so when choosing between large and small circles.

Cats choose the larger amount of food significantly more often than chance in the Control A condition, which is in line with previous findings [13,23,41]. In trials of experimental condition, however, they did not differentiate between the different shapes. This finding is similar to the results of McAllister and Berman [45], Smith [46] and Pisa & Agrillo [23]: even with extensive training, it was difficult for the animals to reach the criteria to discriminate between different shapes with the same surface area, with either food or artificial pictures as stimuli.

The cats' choices might be explained by the species' ecological background. Cats are solitary hunters, thus judging the size of the prey is probably one of the most important tasks for them during the hunt. Prey of different sizes have different costs and benefits, and larger prey can be too costly to be worth hunting (e.g., increased risk of injury, hard to transport, or energetically too costly to handle) [13,47,48]. Cats are also generalist predators, meaning they hunt a wide range of prey from insects to mammals that present different shapes, which might explain the indiscriminate choice in trials of the experimental condition [49,50]. An alternative possibility was that individuals differ in their shape choice, however, the number of trials conducted was insufficient to test this reliably. Since our question concerned cats as a group in general, another type of experiment would be suitable for testing this, either using training or a larger number of repetitions [23,45,46].

We found the opposite phenomenon with dogs, as they showed no significant preference for the larger amount of food either in Control A or Control B conditions. Previous studies show that dogs spontaneously discriminate between food amounts below the ratio of 0.67 [24,35,51] but we were not able to replicate these results. One explanation is that hunger may have not provided enough motivation to choose the larger portion as a) they are well fed and b) could quickly learn that, no matter the outcome, food would always be available at the end of all trials [see similar problem in 52]. Another possible explanation comes from experiments involving training, in which some dogs never reach the criterion of quantity discrimination required to be tested [51,53,54], suggesting that even repeatedly associating the correct choice to a reward is not enough for some dogs to consistently choose the larger food portions.

The preference for the circle over the crown suggests that what is important for dogs is not the surface area but rather the shape or numerosity of the food stimuli. This might be explained by the dog's lifestyle and ecological background as a pack-living animal; it has been observed that free-ranging dogs live in relatively stable packs that present cooperative scavenging and child-rearing, as well as a dominance hierarchy that results in aggression, competition, and "stealing" behaviors when in the presence of food [55–57]. In this social structure, it is easier to protect one big piece of food from other conspecifics and quickly consume it as compared to many small pieces. This is supported by previous findings, which show that dominant dogs will monopolize a large food source (i.e., a carcass) by remaining close to it and defending it aggressively from other dogs in the pack, thus impeding co-feeding [58]. On the other hand, when conducting dyadic tests with a more limited, yet more numerous, amount of food (10 meat chunks and pieces of dry dog food), the dogs demonstrated peaceful sharing and co-feeding in 97% of trials. Further studies investigating the effects of large vs small food sources in packs and dyads are needed to understand monopolization behaviors. Size of individual items has also been observed to be a determining factor on the choice of chimpanzees, who prefer the option with the largest item even when it has a smaller total amount of food [31].

The lack of effects of age, sex, food type or lifestyle in cats, age, weight, or training in dogs, and age or sex in humans, indicates a relatively homogeneous performance within the samples. However, we found a significant difference between male and female dogs, with females choosing the larger portion of food more often in the control condition, particularly in Control A where the ratio between stimuli was 0.5. The difference in preference observed between male and female dogs has not been shown before.

In a previous study, Normando et al. [43] found that intact male dogs choose the larger amount of food more often than neutered ones. They argued that intact dogs have higher energy requirements then the neutered dogs, thus intact males might concentrate more on tasks involving food. The neutered and intact dogs in our sample did not show significant differences in performance, although this could be due to having a limited sample of intact dogs. Additionally, Normando et al. [43] did not find the same differences within female dogs or between the sexes. We agree with their argument that this difference probably reflects calmness, motivation or attention rather than cognitive ability. It might be possible that some of the female dogs we tested were more attentive or motivated than the male dogs, but any notable differences are difficult to observe in our sample because of its small size, especially after dividing by sex.

Seemingly, humans were able to choose the larger stimuli in the Control conditions with ease. This is not surprising since children can already discriminate between stimuli with 0.9 ratios once they're six years old, and this ability only

keeps improving with age [59]. In the trials of the experimental condition, we found no difference in their choice between circle and crown (eight small pieces), but they preferred the circle over the four dots. The lack of a difference between the stimuli with the largest individual area and the stimuli with the highest numerosity (crown) suggests that humans rely equally on continuous magnitudes and numerosity in their quantity judgement, but the choice of the circle over the four dots indicates that individual surface area might be a more salient property when numerosities are small. These preferences might be linked to the object file and analog magnitude system theory [60]. When below the number of three or four, items can be tracked easily and individually, thus they are treated as discrete events. Above this number, their numerosity is judged by their ratios and their continuous magnitudes. We can assume that visual quantity discrimination depends on the simultaneous analysis of numerical and continuous magnitude cues and, since these correlate and interact, thus they must be processed comprehensively, as proposed by Leibovich et al. [61]. However, the number and size of the elements and their ratio change which system is activated. This brings us back to the number sense and magnitude sense theories. While our findings can neither confirm nor reject any of these theories, they support the notion that humans can use both numbers and magnitude to judge quantity.

Finally, we note that the donut shape was the less frequently chosen shape compared to any of the others. The blank area in the middle of the circle may dissuade participants from choosing it, causing it to look like a lesser amount of area or just making it more difficult to judge, therefore compelling participants to default to any other option. It is also possible that the white circle in the middle of the donut interacts with the black edge to create a 'visual illusion', giving the false perception of a lesser area than a filled black circle (circle shape). This could be related to the Oppel-Kundt or filled space illusion, in which spatial intervals that are filled with visual elements are perceived as greater or longer than unfilled intervals of the same length [62]. In our case, the empty area inside the donut could be underestimated by a similar mechanism, resulting in the impression of a smaller diameter (and thus a smaller area) when the circle is unfilled.

Regarding latencies, for humans we found that they were faster in control trials than in the experimental trials, suggesting that the latter were harder choices and required more complex processing of stimuli [36]. They also made quicker decisions when the control trials involved comparing circles of different areas, suggesting a judgment based on individual surface area of items was faster when there was no interaction with numerosity, which is in accordance with the default choices of toddlers [20,63,64] and with the slower processing of numerosity as compared to the processing of shape that has been observed in the brain activity of adults [5,6].

Our study has several limitations. First, we must acknowledge that the convex hull of the shapes is different. We designed our shapes with the goal that the individual pieces should be well recognized, so we could not experimentally control for this variable; the resulting convex hull sizes are opposite of the numerosity of the shapes, with the circle having the smallest one, followed by the donut, then the dots, and with the crown having the largest. However, the results suggest that none of the species were influenced by the convex hull in their decision making, since they showed some preferences for the shape with the smallest convex hull (circle over dots and circle over donut in the case of humans, circle over crown in the case of dogs), but also for the shapes with the largest hulls (dots over donut and crown over donut in the case of humans). Thus, it does not appear that the convex hull was a continuous magnitude which guided their choices.

The second limitation is the choice of methodology, which allowed us to test different species but makes it difficult to compare the results between humans and non-human animals. Comparative studies, especially when they involve human participants, present difficulties in how to control for difference in motivation, due to a lack of reward, or the spontaneous nature of the task, since instructions are provided. In the present study the stimuli itself were also different, a 3D edible one for the animals and the 2D picture for the humans. One potential direction for future studies involving human participants is to use shapes based on food or other more natural contexts, to provide edible or monetary rewards, or remove the instructions and observe their choice in a more spontaneous manner. A second possibility is to use stimulus different from food with the animals.

A third limitation is regarding the rewards given to the animals during the trials. Allowing the animals to eat the chosen stimuli may have reinforced their perception of a wrong or right choice. However, we found no effect on the order of the trials, suggesting that, at least for the time window of the task, there was no learning effect.

Evidence from the three species suggests that quantity estimation is deeply tied to the ecological context of each species. Future work should focus on testing quantity estimation in different contexts, involving different sources of motivation.

## Supporting information

**S1 File. Instructions for human participants.**
(DOCX)

## Acknowledgments

We thank Carolina Rojas for excellent technical and bibliographical assistance, and all cat and dog owners for allowing us repeated access to their homes and pets and to human participants for providing their time. Also, we appreciate the insight of two very thorough reviewers who greatly improved our manuscript.

## Author contributions

**Conceptualization:** Péter Szenczi, Marcos Rosetti, Oxána Bánszegi.

**Data curation:** Mireia Solé Pi.

**Formal analysis:** Mireia Solé Pi, Péter Szenczi, Marcos Rosetti.

**Funding acquisition:** Oxána Bánszegi.

**Investigation:** Luz A. Espino, Marcos Rosetti, Oxána Bánszegi.

**Methodology:** Mireia Solé Pi, Luz A. Espino, Marcos Rosetti, Oxána Bánszegi.

**Project administration:** Oxána Bánszegi.

**Resources:** Luz A. Espino.

**Software:** Péter Szenczi.

**Supervision:** Péter Szenczi, Marcos Rosetti, Oxána Bánszegi.

**Writing – original draft:** Mireia Solé Pi, Luz A. Espino, Péter Szenczi, Marcos Rosetti, Oxána Bánszegi.

**Writing – review & editing:** Mireia Solé Pi, Marcos Rosetti, Oxána Bánszegi.

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
