## [Decision Letter · Decision Letter 0]

29 Jan 2025

PONE-D-24-40472Continuous or discrete magnitudes? A comparative study between cats, dogs and humansPLOS ONE

Dear Dr. Rosetti,

Thank you for submitting your manuscript to PLOS ONE. After careful consideration, we feel that it has merit but does not fully meet PLOS ONE’s publication criteria as it currently stands. Therefore, we invite you to submit a revised version of the manuscript that addresses the points raised during the review process. Your manuscript has been thoroughly reviewed by two independent reviewers who are both of the opinion that the work has merit but that further revision is required. Please see their detailed reviews below.

We look forward to receiving your revised manuscript.

Kind regards,

I Anna S Olsson, Ph.D.

Academic Editor

PLOS ONE

Reviewers' comments:

Reviewer's Responses to Questions

**Comments to the Author**

1. Is the manuscript technically sound, and do the data support the conclusions?

Reviewer #1: Partly

Reviewer #2: Yes

2. Has the statistical analysis been performed appropriately and rigorously? 

Reviewer #1: Yes

Reviewer #2: Yes

3. Have the authors made all data underlying the findings in their manuscript fully available?

Reviewer #1: Yes

Reviewer #2: Yes

4. Is the manuscript presented in an intelligible fashion and written in standard English?

Reviewer #1: Yes

Reviewer #2: Yes

5. Review Comments to the Author

Reviewer #1: The manuscript presents a set of experiments using somewhat analogue procedures across three different species (cats, dogs and humans) and aimed to test whether subject’s choices in a spontaneous choice paradigm were mainly controlled by numerosity or surface area of the stimuli. Each subject was tested with three different trial types that the authors named Control Condition A – the two stimuli had the same shape but one had a larger amount of food than the other (ratio = 0.5) –, Control Condition B – similar to A but the ratio was equal to 0.67 – and Test Condition – the two stimuli had the same food amount but arranged in different shapes (four different possible arrangements). I think the research question and the experiments are interesting and I congratulate the authors for their attempt to compare different species. First, I will provide some general comments and then, more specific comments and suggestions.

General Comments:

- The introduction and discussion sections appear to diverge on their focus. The introduction gives a great emphasis on theories and studies addressing does theories thus, leading the reader to expect that the aim of the study was to contrast the theories. I suggest to focus the introduction on previous studies with cats, dogs and humans that try to address the question of control by different stimuli properties – for instance, it would be appropriate to describe the studies you refer in the methods’ section. Moreover, one advantage of this approach is that it would prepare the reader to for your procedure.

- To improve clarity, it might be helpful to reorganize the methods’ section. Nevertheless, I believe that if you address the first point, it will also help.

- Figure 2 illustrates the procedure in a really nice way. I was thinking that it may be helpful to make it appear earlier in the manuscript.

- In the same line, I think the results’ section would be easier to follow if you divide it by species and not by conditions. In case you maintain it by conditions, use the same names you used before.

- Consider identifying the test you are using when describing the results. Something like “Binominal tests were conducted to compare… and showed that…”.

- Discussion and Conclusion do not need to be separate sections. You can name it, for instance as “Conclusion and Discussion” or just “Discussion” and end it with a conclusion.

- Be consistent in terminology. For example, use either numerosity or numerousness.

- Be careful when using “test”, “testing session” and “testing procedure” to describe an entire experimental session/ procedure. Because you used the name test for one of the conditions, at the first, it seems that each condition took place in a different session and testing sessions correspond only to the sessions where animals were exposed to the test condition.

- Do not use the word “random” to describe choice. It is more informative to say that choice was at chance level or did not differ from chance. Notice that animals might be indifferent between the two options because for them, their value is the same. This result would be similar to chance but is not it is not random.

- The clarity of the data analysis could be enhanced by rephrasing it.

Specific Comments:

Abstract

- The first sentence is too general. You can say something like: “A long-standing question in the study of quantity discrimination is what stimulus properties are controlling choice. While some species….”.

- The sentence “dogs did not differentiate between either of these ratios” does not make much sense to me. I believe you want to state that dogs did not differentiate between the two options (smaller vs. larger amount of food) regardless the ratio between them.

- Keywords: add the species, maybe spontaneous choice; Remove those that are not the focus of the study (e.g., last resort hypothesis).

Introduction

- If the focus remains on the theories (see general comment), they should be better explained providing references for each of them and behavioral studies that test them should be described.

- Provide specific examples in cases as this (p.3, line 56): “certain magnitudes (…), others…”.

- p. 3, line 74: substitute “later on” for “from now on”

- p.4, line 84: studies test animals’ preferences through choice and not animals’ choice.

- p. 5, first paragraph: Maintain the order of the species across sentences: first cats and then, dogs (as in the first sentence).

- p.5, line 97: What you described as Weber’s law is only part of it and it is usually known as the distance effect – the greater the distance between two stimuli, the easier it is to discriminate between them (e.g., it is easier to discriminate between 2 and 6 than between 2 and 3). The Weber’s law combines this distance effect with a size effect and states that discrimination is ratio-dependent: pairs of numerical stimuli with the same ratio are equally easy to discriminate. As an example, discriminability is equal between 2 and 6 and between 3 and 9 because the two pairs have the same ratio (r = 3). Moreover, the greater the ratio the easier it is to discriminate between a pair. That is, it is easier to discriminate between 3 and 12 (r = 4) than between the pairs in the previous example.

- p. 5, line 102: Go slower and give more details on studies that are central to your own study.

- p.5, line 113: use “vs.” instead of “or”

- p.6, line 117: use “choice” instead of “decision”; substitute “we included humans” by something like “we also conducted an experiment with humans”.

Method

- Regarding the structure of the Methods’ section (see general comment above), I found it difficult to understand the experimental setup without knowing what was the task of the animals/humans. I am not completely convinced that my suggestion is the better option (see below), it is possible that you may find a better one.

o Subjects (it is more appropriate to use subjects when referring to non-humans).

Cats and Dogs

Humans

o Procedure (join the description of the stimuli within the description of the task and how data collection took place).

Cats and Dogs

Humans

o Data Analysis

- I suggest the use of numerical digits, in particular when accompanied by a unit (e.g., kg, hr, s).

- p.6, line 134 – 136: I would recommend to be more specific: What behaviors were observed in each species that signaled that the animals were “nervous/anxious”? How did you measure a lack of motivation (e.g., not approaching food?)? Also, substitute “occasions of testing” by trials.

- p. 7, line 138 – 140: Make sure that the criteria used is well described. Suggestion: “Data from 3 cats and 6 dogs was not included in the data analysis because the animals showed a lateral bias. That is, they choose the option presented in one side (left or right) in at least five out of y trials on three consecutive testing sessions or in at least 11 out of 14 trials in one session.”

- p. 7, lines 141 to 148: this information can be presented in a table.

- p. 7, line 152: how did you recruited the human participants? Email? At classes?

- p. 7, line 154: change “be between 18 and 26 years old” to “have between…”

- p. 7, line 155: You should add information regarding how did you assessed if the inclusion criteria was met (online questionnaire? Or other?).

- In my opinion, the excluded participants should go at the beginning of the results’ section. In particular, when excluded after data collection.

- p. 7, line 159: As it is, the mean age seems to refer only to the women participants. Consider also referring the minimum and maximum age.

- p. 8, line 166: I think there is probably a mistake in the first two sentences because they are in disagreement. The authors started by saying that they consider the type of reinforcement important. But the next sentence says that previous studies found the type of food had no effect on performance. Taken the two sentences together, the following sentence (line 168) does not make sense.

- p. 8, line 169: Describe in greater detail the procedure to test what was the preferred food. Was it a three-choice? Did you equal the amount? Were the three plates in the same order for all cats?

- p.8, line 171: Data collection involved 28 and not 25 cats since the other 3 were only excluded due to a lateral bias during data analysis.

- p.8, lines 177 – 180: This paragraph is confusing. Are tasks and conditions the same thing? Maybe this information would be easier to understand if it appeared after explaining the task. And, at that point, explain each condition separately and what was its purpose. We can also say something like “Each animal completed one experimental 14-trial session that comprised three different trial types: control A, control B and test trials.” Then, proceed explaining details on each trial type.

- p.9, line 184: Refer to the Figure before starting its description and use the letters in the text (i.e., A instead of 1, B instead of 3, etc…). Make sure each stimulus is named the same in the text and in the caption of the figure.

- p. 10, line 216: For replication purposes, I think it is important to make the exact instructions given to human participants available (for instance, as an appendix or even in the manuscript).

- p. 12, line 259: I suggest saying “…to avoid the use of environmental cues.” Also, when using example (e.g.,), there is no need for the use of “etc…” at the end.

- p. 13: Figure 2 shows the experimental setup and not the testing setup (again, it suggests you are referring to the test condition). The details in the caption of the figure should go on the text.

Results

- While the text refers to percentages, you use proportions in the graphs. I suggest you to use the same in both, either percentages or proportions.

- For the binomial tests, it might be useful to report the 95% Confidence Interval.

- In case of a significant Wilcoxon test, you should report the effect size. It is more conventional to use W instead of V for the test value.

- Use only two decimals for p-values. If p is less than .001, just report it as p < .001.

- p. 334, line 318: in this task there is no correct choice. Thus, you should refer to percentage of choosing the larger (see also, line 328).

- p. 16, line 325: Why did you report the humans’ results as proportion of smaller choices? It is confusing given that you started the sentence saying you compared choice of the larger shape to chance levels, that you reported preference for larger for cats and dogs and, that the figure uses proportion of choice of the larger stimuli.

- Figure 3: First, the authors should explain in the text what the figure shows. Second, the boxplots of each condition should be further apart because there is overlap of individual datapoints between conditions at 0.50. The mean should be observable in the graph since it is what you report in the text. It might be out of ignorance but, what is the black dot inside a grey circle representing?

- p. 17: report the complete statistics.

- Figure 4: The y-axis of graph c) is different than the others. I would eliminate the asterisks from the graphs but I leave it to the authors.

- p.19, line 356: report the statistics. Maybe the mean latency and standard deviation could be combined in a Table or graphically with each condition and species. The standard deviation of cats for Control A made me wonder whether you consider all latencies or eliminated outliers. I was also surprised that cats’ latencies were almost twice those of dogs, do you have any idea why?

- p. 19, line 359: Start with the control conditions as you did in the rest of the results. You should clearly state that you combined the two control conditions.

Discussion

- I recommend starting the discussion by remembering the reader of your goal and a summary of what you did.

- p. 20, line 377: which ratio? Are you talking about control conditions or test? Be more specific.

- In line with the difference you observed between male and female dogs, I was wondering if the restriction could have affected cats and dogs differently because of the difference in their weights. All animals were deprived from food the same amount of time regardless their weight. Also, did you control or have any information regarding the animals’ eating routines? That is, did they usually had food always available or did their owners usually give them food at a particular time of the day and a particular amount per day? I wonder if that may affect how animals responded in the task and their motivation.

- It would be nice to integrate preference results with latencies – are they in accordance?

- p. 22, line 439: eliminate “hollow circle”.

- P. 23, lines 439 to 442: Is it possible that the donut shape creates an illusion regarding amount?

- p. 23, line 442 to 446: I felt it was a big jump. Either eliminate this information or make sure you explain why this is important and how is it related to the results.

- Did you find any effect of presentation order? Is it possible that the animals learned that they would eat the same regardless what option they chose and, thus, at latter trials start to be indifferent?

- I suggest a greater focus and discussion on the differences between the animals and humans’ task: 1) the animals were presented with 3D stimuli and the humans with 2D; 2) animals were reinforced for choosing and the stimuli were primary reinforcers whereas the humans were not. 3) Humans were instructed to choose the larger amount thus the task was not analogue. It would be interesting to make humans complete the task without instructions (as the animals).

I hope you find these comments useful.

Reviewer #2: The current study aims at disentangling whether three different species (cats, dogs, and humans) have a preference towards discriminating stimuli based on total amount of said stimuli (by responding to changes differences in a continuous variable; in this case, surface area) or rather on numerosity (by comparing them on the basis of the number of discrete units of the relevant stimuli). In order to do this, the authors employ a paradigm through which the subjects have first to discriminate between two stimuli with the same shape but a different surface area (first with a difference in total area with a ratio of 0.5 and later with a ratio of 0.67), but later on are presented with two stimuli with the same surface area but presented in a different array —most critical to the purposes of this study, in the form of 1, 4, or 8 discrete units. The stimuli presented come in the form of food for the non-human animals and in the form of different figures in a computer screen for humans. Results show that dogs did not show a preference for the larger surface area in the control sessions, humans did, and cats did only do so with the largest different in ratio (0.5). When they compared stimuli of the same surface area, humans seemed to have a bias against a donut-shaped stimulus and dogs did prefer a single stimulus over an array of 8 discrete units, but cats showed no apparent preference for or against any of the arrays of stimuli used.

I believe the current study tackles one of the major confounds when it comes to the study of quantity discrimination, as many of the existing studies in the literature do often equate the discrimination between "numbers" (i.e., amount of discrete units of a given stimulus) to the discrimination between continuous variables. However, I do believe this study does have some issues —both in the methodological and reporting departments— that should be adressed.

What I do believe to be the most important issue in the current study (and the authors briefly touch on it, but not to the degree they ought, in my opinion) is the fact that the existing literature does seem to provide enough evidence to consider that the discrimination between quantities of food or neutral stimuli often provide starkly different results. The most glaring example that comes to mind is Macpherson & Roberts (2013) (doi: 10.1016/j.lmot.2013.04.002), where dogs failed to discriminate between different quantities of food (other than the most simple 1 piece of food vs. 0 pieces), but later on a single dog performed a pilot in which they were able to distinguish ratios of up to 0.75. This is also found (albeit to a lesser degree) in Range et al. (2014) (doi: 10.3389/fpsyg.2014.01299), where dogs were only able to discriminate pieces of food up to a ratio of 0.5. To contrast this one, Rivas-Blanco et al. (number 36 in your citations) showed that they were able to discriminate up to a ratio of 0.8 when it came to dots on a computer screen. So, at the very least for dogs, this difference in paradigm used (quantity discrimination between pieces of food or other neutral stimuli) can potentially impact the outcome of the experiment drastically.

The authors do indeed mention differences between "spontaneous" and "trained" quantity discrimination abilities in the introduction, but rather, what seems to me to be the greatest difference between the both is something they do mention later on in the discussion: whenever the subjects are required to differenciate between two quantities of food they are always rewarded, but this is not the case if they are asked to differentiate between two neutral stimuli. As a result, considering the fact that subjects in the study may not be particularly hungry (the authors did ask the owners of cats and dogs to not feed their animals for 4 hours prior to the study, but in my experience, at least dogs are usually fed twice a day, which would mean the animals may have not been food-motivated enough), figuring out the method to obtain the higher amount of food may have not been a priority for the animals, and therefore, the non-human subjects may have not been motivated enough to choose the larger amount of stimuli, regardless of whether or not they had the capability to make this discrimination (this seems to be particularly the case in dogs, but it may have also affected cats' performance to a certain degree as well).

As such, I believe the authors should consider the task they presented to the humans and to the non-human animals as two different experiments, and both the discussion and the statistical analyses should be separate. This last point is particularly important, since humans did not only do a different task with different affordances, but also received explicit text-based instructions that the animals (of course) lacked. Therefore, there are three degrees of separation between humans and non-human animals: 1) their species, 2) the task used, 3) the information provided; which I believe should make statistical analyses comparing all three populations inadequate. On this topic, the authors do mention the possibility of testing humans with food or money instead of shapes, but another possibility I reckon worth considering is going the opposite route and using the computer program for all three species (for example, through the use of touch screens). By presenting all three species with the same computer program and having them learn the contingencies of the task through training, it would be possible to equate the participants' information on the task (as well as providing a method to test the non-human animals in this paradigm that does not require the use of language).

Another comment related to statistics, I found it strange that there was no comparison between the performance with each shape in the control conditions. It could be the case that the subjects had different performances based on the shape used, which may or may not be linked to their preferences in the test phase. This is especially relevant in the case of non-human animals, as them struggling (to a higher or lesser degree) to discriminate between the different areas could be partly due to them having issues with one of the shapes in particular.

Other than this, I found there were some important absences in the way the methodology is reported that preclude (or at least, greatly difficult) a possible replication of the described paradigms. Most notoriously, I believe it would be very important to provide the information the human participants received through the program, as this may have greatly influenced their perception of the task at hand. Other feature of the experiment that I believe requires further reporting is the arrays of stimuli used in the test phase. It is my understanding that the "circle", "dots", and "crown" shapes are meant to represent different number of stimuli (1, 4, and 8, respectively), but I reckon this should be explained clearly in the text. On that topic, I do not understand the inclusion of the "donut" shape, as this would technically represent the same number of discrete stimuli (1) as the "circle" configuration. Not to say it serves no purpose, but the reasoning behind its inclusion shoud be clearly explained in the text. On a minor note concerning the reporting and replicability of the study: the apparent specificity of the areas used for the stimuli (i.e., the fact that the authors did not opt for "round numbers") leads me to believe there may be a practical reason for choosing these areas in particular (e.g., the radius of the circles do happen to be a round number, or this does align with the size constrains of the plates used or the computer screen). If this is the case, this too should be reported.

Additionally, I believe the authors would benefit from making a more throughout review of the existing literature on the topic. Some absences I found worth mentioning are the aforementioned Macpherson & Roberts (2013) paper, and Agrillo et al. (2012) (doi: 10.1371/journal.pone.0031923), which discusses that although humans do make quantity discrimination assessments perfectly (or close to), their response time does slow down as the ratio between the quantities presented gets smaller (which I believe goes along the lines of the reported differences in response time in humans between the control and test phases; as the difficulty of the task increases, they do require more time to make their choice).

Finally, one minor style comment, but the authors should consider the way they report numbers. Some sentences do include numbers both as text and as numerals (with a glaring example being line 141-144). On that topic, to my knowledge, when used with abbreviations, numbers are always presented as numerals (e.g., line 130: it should be 10 kg, not ten kg).

Line comments:

Line 56: Consider starting a new paragraph.

Lines 73-74: I believe "(animals later on)" could be safely removed.

Line 76: I would suggest a semicolon after "surface area", for clarity's sake.

Line 83 & ff.: Consider re-structuring the introduction. The "last resort" hypothesis is mentioned in the paragraph starting with line 63, then there is a paragraph in between, and then this idea is explored again. Additionally, the way I see it, the main topic of the paragraph is that some animals employ discrete numbers while others seemingly do not. Therefore, I would consider re-structuring the paragraph so that it starts exploring that idea.

Line 87: Rivas-Blanco et al. 2020 (citation number 36 in your paper) could arguably also be included when you mention dogs here. Dogs were provided with numerical data, but the controls suggest that they made their comparisons based on continuous information.

Line 93: The change is topic is rather drastic, consider providing a segue.

Line 97-98: I feel the topic of Weber's law should be introduced earlier.

Line 102 & ff.: Same as before, I should consider restructuring the introduction so that the ideas discussed flow naturally into each other. The introduction is rather short, so there should be no need of a paragraph starting with "as mentioned above" or any variation thereof by carefully paying attention to the structure.

Line 119: A brief overview of the conditions used and the reasoning behind them before the methods would prove very useful for the reader.

Line 130: Please provide reasoning for the weight requirements for the dogs. Also "in the case of dogs, be over ten kg" > "in the case of dogs, they had to be over 10 kg".

Line 131: The proper abbreviation for "hour" is "h".

Line 136: "testing sessions" rather than "occasions of testing"?

Line 141 & ff.: It would be really useful if the authors could provide a subject list with some basic information for the subjects (age, sex, breed in the case of dogs...).

Line 153: "a) they should" > "they should a)".

Line 166: While I agree on the notion, this seems like an odd way to start the paragraph.

Lines 167-168: I do not think I understand the authors' reasoning. If the authors really concur with the notion that the type of food used for testing has no effect on cats' performance, then why would it be necessary to check their preferences.

Lines 177 & ff.: I reckon more details should be provided in this paragraph.

Line 182: Maybe "quantity discrimination abilities" rather than "ability of quantity discrimination"?

Line 191: Consider starting a new sentence instead of using a semicolon.

Lines 200-201: However, the dogs were not provided with chairs to account for this. Care to elaborate?

Lines 218-220: This should be moved to the cats and dogs' section. It is the same for all species, but it would prove more useful over there (since it comes earlier in the paper).

Lines 221-222: Consider mentioning that these controls are meant to be similar to the ones used for cats and dogs. I reckon it makes it easier for the reader.

Line 227: I reckon it would be easier if the paragraph started by emphasizing the fact that this was done exclusively in humans (e.g., rather than "for thethe human participants", something like "as opposed to dogs and cats", or something to that effect. Additionally, it would be interesting to know the reasoning behind this additional control.

Line 240: The 4-hours food deprivation was brought elsewhere. I do not have a particular preference about the location of that snippet of information, but I reckon it should be kept in only one place.

Line 245: May be worthwhile to further explain the "gently hold it there from behind"; not sure if it is completely clear.

Line 272: Consider rewording, this makes it sound like the dogs were also on a chair.

Line 288: Maybe "all analysis were" rather than "all analyses was"

Line 288 & ff.: On the models below you mention having checked for the potential influence of random effects, but here you add together data from all participants, which could lead to pseudo-replication issues. Why?

Line 296 & ff.: It would be useful to provide the formula of the resulting model.

Line 301: Remove one of the two periods.

Line 304-306: Not sure if I understand what this means. Care to elaborate?

Line 307-309: Same as above, this does not account for differences within each individual.

Line 330-331: Consider rephrasing: the program failed to record, not the glitch.

Lines 362-363: Please specify what is meant by "comparing differently sized circles".

Line 383-383: As mentioned above, please specify what is meant with "circles of different areas".

Line 387: Consider removing "their test".

Line 406 & ff.: Only one of the combinations was significant. That makes it more likely that we may be facing a spurious correlation. I think this should be brought up in the text.

Lines 424-426: Would not an equal reliance on both factors imply that they would prefer a higher number of discrete forms in the screen? That is to say, given that the participants were (as far as I understand) instructed to pick the largest option, and considering that in the test session both stimuli had the same surface area, the only distinguishing factor remaining would be number. Or put in other words: if there are no differences in surface area (and thus discriminating on the basis of continuous variables was not possible), the only other relevant variable would be number.

Line 436: I think it would be better to start a new sentence instead of using a semicolon.

Line 439-446: I am not sure how much value this provides to the paper.

Line 447: Comma after "humans".

Lines 458-460: As I mentioned in the beginning, given that this is the case, I reckon it is not justified to put all species in the same model.

Line 469 & ff.: I would try to tone down the claims. I am not sure whether this could be said with the results of the current study alone.

6. PLOS authors have the option to publish the peer review history of their article (what does this mean? ). If published, this will include your full peer review and any attached files.

**Do you want your identity to be public for this peer review?** For information about this choice, including consent withdrawal, please see our Privacy Policy .

Reviewer #1: No

Reviewer #2: **Yes: ** Dániel Rivas-Blanco

---

## [Author Response · Author response to Decision Letter 1]

27 Feb 2025

5. Review Comments to the Author

Reviewer #1: The manuscript presents a set of experiments using somewhat analogue procedures across three different species (cats, dogs and humans) and aimed to test whether subject’s choices in a spontaneous choice paradigm were mainly controlled by numerosity or surface area of the stimuli. Each subject was tested with three different trial types that the authors named Control Condition A – the two stimuli had the same shape but one had a larger amount of food than the other (ratio = 0.5) –, Control Condition B – similar to A but the ratio was equal to 0.67 – and Test Condition – the two stimuli had the same food amount but arranged in different shapes (four different possible arrangements). I think the research question and the experiments are interesting and I congratulate the authors for their attempt to compare different species. First, I will provide some general comments and then, more specific comments and suggestions.

General Comments:

- The introduction and discussion sections appear to diverge on their focus. The introduction gives a great emphasis on theories and studies addressing does theories thus, leading the reader to expect that the aim of the study was to contrast the theories. I suggest to focus the introduction on previous studies with cats, dogs and humans that try to address the question of control by different stimuli properties – for instance, it would be appropriate to describe the studies you refer in the methods’ section. Moreover, one advantage of this approach is that it would prepare the reader to for your procedure.

ANSWER: WE HAVE RESTRUCTURED THE INTRODUCTION AND PUT MORE EMPHASIS ON PREVIOUS STUDIES (LINES 64-73).

- To improve clarity, it might be helpful to reorganize the methods’ section. Nevertheless, I believe that if you address the first point, it will also help.

ANSWER: WE REORGANIZED THE SECTION AS THE REVIEWER SUGGESTED BELOW. PLEASE SEE METHODS SECTION.

- Figure 2 illustrates the procedure in a really nice way. I was thinking that it may be helpful to make it appear earlier in the manuscript.

ANSWER: DONE. LINES 170.

- In the same line, I think the results section would be easier to follow if you divide it by species and not by conditions. In case you maintain it by conditions, use the same names you used before.

ANSWER: WE WOULD LIKE TO KEEP THE DESCRIPTION OF THE RESULT SECTION IN THIS WAY BECAUSE WE THINK IT IS EASIER FOR THE READERS TO FOLLOW AND COMPARE THE PERFORMANCE OF THE SPECIES IN THE DIFFERENT CONDITIONS.

- Consider identifying the test you are using when describing the results. Something like “Binominal tests were conducted to compare… and showed that…”.

ANSWER: WE HAVE ADDED THE NAME OF THE TESTS WHEN DESCRIBING THE RESULTS, WE HOPE THIS IMPROVES THE CLARITY (LINES 334, 337, 339, 343, 346, 349, 352, 356).

- Discussion and Conclusion do not need to be separate sections. You can name it, for instance as “Conclusion and Discussion” or just “Discussion” and end it with a conclusion.

ANSWER: WE HAVE REMOVED THE “CONCLUSION” HEADER AND MERGED BOTH SECTIONS.

- Be consistent in terminology. For example, use either numerosity or numerousness.

ANSWER: THANK YOU FOR YOUR SUGGESTION. WE HAVE CHOSEN TO USE NUMEROSITY CONSISTENTLY.

- Be careful when using “test”, “testing session” and “testing procedure” to describe an entire experimental session/ procedure. Because you used the name test for one of the conditions, at the first, it seems that each condition took place in a different session and testing sessions correspond only to the sessions where animals were exposed to the test condition.

ANSWER: WE HAVE MODIFIED THE TERMS TO CLARIFY THEIR MEANINGS AND USE DISTINCT TERMS TO REFER TO EACH OF THE CATEGORIES MENTIONED BY THE REVIEWER.

- Do not use the word “random” to describe choice. It is more informative to say that choice was at chance level or did not differ from chance. Notice that animals might be indifferent between the two options because for them, their value is the same. This result would be similar to chance but is not it is not random.

ANSWER: THE REVIEWER IS CORRECT. WE HAVE REPLACED INSTANCES OF RANDOM ACCORDINGLY (LINES 335, 337, 344, 350 AND 352)

- The clarity of the data analysis could be enhanced by rephrasing it.

ANSWER: THE DATA ANALYSIS HAS BEEN REPHRASED, WE HOPE THIS IMPROVES ITS CLARITY (LINES 290-325).

Specific Comments:

Abstract

- The first sentence is too general. You can say something like: “A long-standing question in the study of quantity discrimination is what stimulus properties are controlling choice. While some species….”.

ANSWER: THANK YOU FOR THE SUGGESTION. WE HAVE MODIFIED THE TEXT ACCORDINGLY (LINE 26).

- The sentence “dogs did not differentiate between either of these ratios” does not make much sense to me. I believe you want to state that dogs did not differentiate between the two options (smaller vs. larger amount of food) regardless of the ratio between them.

ANSWER: CORRECT. WE HAVE AMENDED THE SENTENCE (LINES 33-34).

- Keywords: add the species, maybe spontaneous choice; Remove those that are not the focus of the study (e.g., last resort hypothesis).

ANSWER: THANKS FOR THE SUGGESTION. WE HAVE ADDED THE KEYWORDS PROVIDED (LINES 44-45).

Introduction

- If the focus remains on the theories (see general comment), they should be better explained providing references for each of them and behavioral studies that test them should be described.

ANSWER: WE NOW PROVIDE A LONGER DESCRIPTION OF THE BEHAVIORAL STUDIES SUPPORTING THE THEORIES, AND THE CORRESPONDING REFERENCES (LINES 64-73, 94-102)

- Provide specific examples in cases as this (p.3, line 56): “certain magnitudes (…), others…”.

ANSWER: GOOD SUGGESTION. WE HAVE ADDED AN EXAMPLE (LINE 57-58).

- p. 3, line 74: substitute “later on” for “from now on”

ANSWER: DONE (LINE 65).

- p.4, line 84: studies test animals’ preferences through choice and not animals’ choice.

ANSWER: DONE (LINE 85).

- p. 5, first paragraph: Maintain the order of the species across sentences: first cats and then, dogs (as in the first sentence).

ANSWER: CORRECTED.

- p.5, line 97: What you described as Weber’s law is only part of it and it is usually known as the distance effect – the greater the distance between two stimuli, the easier it is to discriminate between them (e.g., it is easier to discriminate between 2 and 6 than between 2 and 3). The Weber’s law combines this distance effect with a size effect and states that discrimination is ratio-dependent: pairs of numerical stimuli with the same ratio are equally easy to discriminate. As an example, discriminability is equal between 2 and 6 and between 3 and 9 because the two pairs have the same ratio (r = 3). Moreover, the greater the ratio the easier it is to discriminate between a pair. That is, it is easier to discriminate between 3 and 12 (r = 4) than between the pairs in the previous example.

ANSWER: WE HAVE OPTED TO REMOVE THE MENTION OF WEBER’S LAW AND KEEP THE POINT OF USING THE RATIO TO DISCRIMINATE IN ORDER TO KEEP THE SENTENCE CLEAR (LINES 98 AND 99)

- p. 5, line 102: Go slower and give more details on studies that are central to your own study.

ANSWER: WE HAVE ADDED SOME DETAIL TO THE MENTIONED STUDIES AND AN ADDITIONAL REFERENCE TO SUPPORT THE BACKGROUND OF THE CURRENT WORK (LINES 94-102).

- p.5, line 113: use “vs.” instead of “or”

ANSWER: THE USE OF “VS” IS NOT APPROPRIATE IN THIS CASE, BUT WE HAVE AMENDED THE SENTENCE (LINE 120).

- p.6, line 117: use “choice” instead of “decision”; substitute “we included humans” by something like “we also conducted an experiment with humans”.

ANSWER: DONE (LINES 121 AND 124)

Method

- Regarding the structure of the Methods’ section (see general comment above), I found it difficult to understand the experimental setup without knowing what was the task of the animals/humans. I am not completely convinced that my suggestion is the better option (see below), it is possible that you may find a better one.

o Subjects (it is more appropriate to use subjects when referring to non-humans).

Cats and Dogs

Humans

o Procedure (join the description of the stimuli within the description of the task and how data collection took place).

Cats and Dogs

Humans

o Data Analysis

ANSWER: WE REORGANIZED THE METHODS SECTION AS SUGGESTED BY THE REVIEWER. WE ALSO DIRECT THE READER'S ATTENTION TO THE FIGURES AT THE BEGINNING TO THE DESCRIPTION WHICH MIGHT HELP TO UNDERSTAND THE EXPERIMENTAL SETUP.

- I suggest the use of numerical digits, in particular when accompanied by a unit (e.g., kg, hr, s).

ANSWER: GREAT SUGGESTION. WE HAVE AMENDED IT ACCORDINGLY THROUGHT THE MS.

- p.6, line 134 – 136: I would recommend to be more specific: What behaviors were observed in each species that signaled that the animals were “nervous/anxious”? How did you measure a lack of motivation (e.g., not approaching food?)? Also, substitute “occasions of testing” by trials.

ANSWER: WE HAVE PROVIDED EXAMPLES OF SIGNALS SHOWING THE ANIMALS NERVOUS/ANXIOUS STATE BUT THINK THAT THE LACK OF MOTIVATION IS CLEAR STATED. WE ALSO REPLACED THE TERM AS SUGGESTED (LINES 145-147)

- p. 7, line 138 – 140: Make sure that the criteria used is well described. Suggestion: “Data from 3 cats and 6 dogs was not included in the data analysis because the animals showed a lateral bias. That is, they choose the option presented in one side (left or right) in at least five out of y trials on three consecutive testing sessions or in at least 11 out of 14 trials in one session.”

ANSWER: WE HAVE ADDED A SENTENCE SIMILAR TO THE REVIEWER’S SUGGESTION (LINES 148-150)

- p. 7, lines 141 to 148: this information can be presented in a table.

ANSWER: WE HAVE ADDED A TABLE TO PRESENT THE DEMOGRAPHIC INFORMATION FOR THE SUBJECTS. SEE TABLE 1, LINE 152.

- p. 7, line 152: how did you recruited the human participants? Email? At classes?

ANSWER: DONE. LINE 158.

- p. 7, line 154: change “be between 18 and 26 years old” to “have between…”

ANSWER: THIS AND OTHER SMALL CORRECTIONS WERE DONE IN THESE LINES (159-160)

- p. 7, line 155: You should add information regarding how did you assessed if the inclusion criteria were met (online questionnaire? Or other?).

ANSWER: DONE. LINES 161.

- In my opinion, the excluded participants should go at the beginning of the results’ section. In particular, when excluded after data collection.

ANSWER: SINCE THE EXCLUDED PARTICIPANTS WERE NOT INCLUDED IN THE DATA ANALYSIS AND THE RESULTS’ SECTION REFERS TO THE FINAL DATA SET, WE WOULD LIKE TO KEEP OUR DESCRIPTION AS IT IS NOW. HOWEVER, WE ADDED AN EXTRA SENTENCE TO THE BEGINNING OF THE RESULT SECTION TO SUMMARIZE THE NUMBER OF ANIMALS WHICH WERE VISITED, TESTED AND INCLUDED IN THE FINAL ANALYSIS (SEE LINES 332-333).

- p. 7, line 159: As it is, the mean age seems to refer only to the women participants. Consider also referring the minimum and maximum age.

ANSWER: THIS INFORMATION IS NOW ON TABLE 1.

- p. 8, line 166: I think there is probably a mistake in the first two sentences because they are in disagreement. The authors started by saying that they consider the type of reinforcement important. But the next sentence says that previous studies found the type of food had no effect on performance. Taken the two sentences together, the following sentence (line 168) does not make sense.

ANSWER: THE REVIEWER IS CORRECT, AND WE HAVE AMENDED THE SENTENCE (LINES 176-180).

- p. 8, line 169: Describe in greater detail the procedure to test what was the preferred food. Was it a three-choice? Did you equal the amount? Were the three plates in the same order for all cats?

ANSWER: WE ADDED DETAIL TO THIS SENTENCE IN LINES 177-182.

- p.8, line 171: Data collection involved 28 and not 25 cats since the other 3 were only excluded due to a lateral bias during data analysis.

ANSWER: WE CORRECTED THIS NUMBER. IT WAS A TYPO (LINE 181)

- p.8, lines 177 – 180: This paragraph is confusing. Are tasks and conditions the same thing? Maybe this information would be easier to understand if it appeared after explaining the task. And, at that point, explain each condition separately and what was its purpose. We can also say something like “Each animal completed one experimental 14-trial session that comprised three different trial types: control A, control B and test trials.” Then, proceed explaining details on each trial type.

ANSWER: AS MENTIONED ABOVE WE MODIFIED THE TERMS TO CLARIFY THEIR MEANINGS AND USE DISTINCT NAMES FOR EACH OF IT. WE ALSO REORGANIZED THE METHODS SECTION AND DIRECT THE READER’S ATTENTION TO THE FIGURES AT THE BEGINNING OF THE DESCRIPTION OF THE PROCEDURES.

- p.9, line 184: Refer to the Figure before starting its description and use the letters in the text (i.e., A instead of 1, B instead of 3, etc…). Make sure each stimulus is named the same in the text and in the caption of the figure.

ANSWER: GREAT SUGGESTION. DONE. SEE LINES 190-192.

- p. 10, line 216: For replication purposes, I think it is important to make the exact instructions given to human participants available (for instance, as an appendix or even in the manuscript).

ANSWER: WE HAVE INCLUDED THE INSTRUCTIONS GIVEN TO THE PARTICIPANTS IN A SUPPLEMENTARY MATERIAL (REFERENCE IN LINE 272-273).

- p. 12, line 259: I suggest saying “…to avoid the use of environmental cues.” Also, when using example (e.g.,), there is no need for the use of “etc…” at the end.

ANSWER: DONE. LINE 256.

- p. 13: Figure 2 shows the experimental setup and not the testing setup (again, it suggests you are referring to the test condition). The details in the caption of the figure should go on the text.

ANSWER: WE HAVE CHANGED TESTING FOR EXPERIMENTAL AND SUMMARIZED THE CAPTION TO AVOID REPEATING INFORMATION THAT WAS ALREADY IN-TEXT. SEE FIGURE 2.

Results

- While the text refers to percentages, you use proportions in the graphs. I suggest you use the same in both, either percentages or proportions.

ANSWER: DONE. SEE LINES 334 AND 343.

- For the binomial tests, it might be useful to report the 95% Confidence Interval.

ANSWER: THANK YOU FOR THE SUGGESTION, WE HAVE ADDED THE CONFIDENCE INTERVAL FOR EACH BINOMIAL TEST. LINES 336-337, 338, 345-346, 352 AND 425.

- In case of a significant Wilcoxon test, you should report the effect size. It is more conventional to use W instead of V for the test value.

ANSWER: WE HAVE REPORTED THE EFFECT SIZE FOR THE SIGNIFICANT WILCOXON TESTS AND ADOPTED THE W NAMING CONVENTION. LINES 348, 399, 400 AND 411.

- Use only two decimals for p-values. If p is less than .001, just report it as p < .001.

ANSWER: WE REPLACED WERE APPROPRIATE (LINE 336), LEFT AS IS WHERE THE VALUE WAS EQUAL TO .001.

- p. 334, line 318: in this task there is no correct choice. Thus, you should refer to percentage of choosing the larger (see also, line 328).

ANSWER: DONE. LINES 340-341.

- p. 16, line 325: Why did you report the humans’ results as proportion of smaller choices? It is confusing given that you started the sentence saying you compared choice of the larger shape to chance levels, that you reported preference for larger for cats and dogs and, that the figure uses proportion of choice of the larger stimuli.

ANSWER: WE CORRECTED THE PROPORTIONS TO REFLECT THE CHOICE OF THE LARGER STIMULUS. LINES 350-357.

- Figure 3: First, the authors should explain in the text what the figure shows. Second, the boxplots of each condition should be further apart because there is overlap of individual datapoints between conditions at 0.50. The mean should be observable in the graph since it is what you report in the text. It might be out of ignorance but, what is the black dot inside a grey circle representing?

ANSWER: WE HAVE SEPARATED THE BOXPLOTS AND REDUCED THE SIZE OF THE DOTS SO THAT THERE IS NO OVERLAP OF INDIVIDUAL DATAPOINTS. WE DO NOT REFER TO THE MEAN IN THE TEXT FOR THIS SECTION, SO WE HAVE DECIDED TO KEEP THE MIDDLE LINE OF OUR

---

## [Decision Letter · Decision Letter 1]

23 Apr 2025

PONE-D-24-40472R1Continuous or discrete magnitudes? A comparative study between cats, dogs and humansPLOS ONE

Dear Dr. Rosetti,

Thank you for submitting your manuscript to PLOS ONE. After careful consideration, we feel that it has merit but does not fully meet PLOS ONE’s publication criteria as it currently stands. Therefore, we invite you to submit a revised version of the manuscript that addresses the points raised during the review process.

**Thank you for submitting a revised version of your manuscript. As outlined in more detail in the reviewer reports below, there are remaining issues that need addressing.**

We look forward to receiving your revised manuscript.

Kind regards,

I Anna S Olsson, Ph.D.

Academic Editor

PLOS ONE

Reviewers' comments:

Reviewer's Responses to Questions

**Comments to the Author**

1. If the authors have adequately addressed your comments raised in a previous round of review and you feel that this manuscript is now acceptable for publication, you may indicate that here to bypass the “Comments to the Author” section, enter your conflict of interest statement in the “Confidential to Editor” section, and submit your "Accept" recommendation.

Reviewer #1: (No Response)

Reviewer #2: (No Response)

2. Is the manuscript technically sound, and do the data support the conclusions?

Reviewer #1: Yes

Reviewer #2: Yes

3. Has the statistical analysis been performed appropriately and rigorously? 

Reviewer #1: Yes

Reviewer #2: I Don't Know

4. Have the authors made all data underlying the findings in their manuscript fully available?

Reviewer #1: No

Reviewer #2: Yes

5. Is the manuscript presented in an intelligible fashion and written in standard English?

Reviewer #1: Yes

Reviewer #2: Yes

6. Review Comments to the Author

**Reviewer #1:**  Great job improving the clarity of the manuscript! Overall, the authors did a careful revision and adequately responded to my previous comments. The procedure is much clearer. However, I still have some comments and suggestions but much minor compared to the previous ones. As a general comment, the authors tend to use sentences and paragraphs that are too big and with too many ideas (this is even more salient in the discussion).

Abstract

p-2, line 31-32: be more specific than “some preference” or, at least, specify a preference for what.

p.2, line 34: at this point, the reader does not know what is the control condition. Instead of using the name of the condition, describe it as you did in the rest of the sentence.

Introduction

p. 4, line 68: processing numerical information involves a larger cognitive load compared to what? Also, what “this strategy” is referring to? You did not describe any strategy in the previous sentence.

p.4, lines 73 to 76: Divide the sentence in key ideas. As it is, it is difficult to understand the message.

p.4, line 82: why do you considered the use of number rather than of the continuous magnitudes a bias?

p.5, line 91: species rely on the ratio between what? the numerosity? the surface area? of the two stimuli. The information inside brackets is important to understand the results so it should be a regular sentence.

p.5, line 97: decompose the sentence, there are too many ideas and information. How did the animals choose (response) and what was the correct response? Choosing the three item-set? End the sentence at this point. Then you can start the next sentence with results. This is only a suggestion:” Cats reliably learned to choose (...). But, when surface area was controlled, rats were indifferent between the two stimuli suggesting their previous preference was based on the surface are and not on quantity.”

p.5, line 100: consider substituting “above 0.5 ratios” for “when the ratio between the two stimuli was greater than 0.5”. In the same sentence, I did not understood what the authors meant with “both numerical and continuous magnitude differences”.

p.5, line 105: Explain what you mean by “counterbalanced” in this case.

p.6, line 114: The clarity of this paragraph can be increased.

Methods

p.6, line 127 to 129: weight at least 10 kg instead of “over 10kg” and the information that comes next should not be part of the same paragraph. Suggestion: “This last constraint serve to….”.

p. 7, line 141: “details of”

Table 1: I believe there is a better way to present the table. Since there is much more information regarding cats and dogs, I would suggest a table for those and the information regarding the humans can either be in a separate, simpler table or in the text. Below there is a table suggestion with only some information just to exemplify. You may need to switch the lines with columns due to the space occupied when more info is added.

Age Sex

M (SD) Range Female Male

Neutered Intact Neutered Intact

Cats 6.4 (3.7) 1 – 14 9 1 12 4

Dogs 4.1 (3.1) 1 – 13 11 4 9 4

p. 10, line 189: the ratio between the surface areas of the two food portions was 0.5

p.14, line 249: the experimenter asked the participant’s initials, age….

p. 14, line 260: what is S1?; “completed trials of three different conditions”

p.14, line 263: “a total of 28 trials, x of each condition”

p.15: Four trials of Control A plus 4 of Control B plus 6 of Experimental do not comprise 28 trials as stated before. I anticipate there were two trial-blocks?!

p.15, line 272: Suggestion: “If participants took more than 10 s to click on either option (i.e., the maximum choice latency allowed), the trial was cancelled and the next trial started immediately.

Results

Instead of “latency of choice”, use “latency to choose”. Be careful when using “female dogs’ choice for the larger” or “human participants’ choice of the larger”, it adds complexity to the information.

Discussion

In general, paragraphs are too long and with too many ideas. Try to divide them.

p.23, line 426: substitute extremely by significantly

p.23, line 428: According to the first part of the sentence, I was expecting the second part to be something like: while cats did not showed a preference for any particular shape or while cats were indifferent between shapes.

P23, line 430: “in the control condition” can be eliminated because you already said it in the beginning of the sentence.

p.25, line 463: did you meant within each sample?.

p.25, line 466: where the ratio between…

p.25, line 466: The difference in preference observed between male and female dogs…

p.25, line 467: Did you check if there was any effect similar to that reported by Simona, Maria?

p.25, line 473: dividing by sex; it might be possible; I think the conclusion of the sentence is odd. I would rather say that any differences may not be possible to observe in your sample because of its small size, mainly when dividing by sex.

p.25, line 476: the discrimination is not between ratios is between two stimuli with a certain ratio between them. Be careful with this among the entire manuscript.

p.25, line 479: please explain what is the result suggests each of the two possibilities. What led you to those possibilities? It seems you are jumping a step in the rationale.

p.26, line 490: …and their ratio change which….

p.26, lines 496 and 497: three repetitions of the word making.

p.26, line 498: I believe words are missing from the sentence.

p.27, line 504: latencies

p.27, line 506 and 507: when the control trials involved circles of different areas

**Reviewer #2: ** General comments

The main issue of this manuscript (and one that remains unchanged in this version) is that the authors do not exercise enough caution when comparing the different species. The paradigm used for both dogs and cats is similar enough to allow for a direct comparison, but humans get a completely different paradigm that involves no food, as well as instructions (that the animals, of course, do not get). Therefore, I think statistical analyses including data from all three species are inapropriate, and comparisons between humans and non-human animals should be kept to a minimum.

The end of the introduction still feels rather bare, I think a paragraph in which the authors discuss the paradigm they used and how this connects to what the experiment is meant to test would be an important addition to the text.

Finally, the authors talk about ecological relevance being the driver behind the methods described. However, dogs are scavengers in nature, and they're well-suited for periods of scarcity. Under the abundance of food experienced by most house pets, optimal foraging (or, in this context, choosing the larger quantity) would arguably rarely be a priority. Particularly if, as the authors mention in the introduction, numerosity happens to be a particularly cognitively loading skill. These considerations do not need to be added to the manuscript, but I still thought they were worth discussing.

In-line comments

103-107: Consider restructuring this sentence, it is somewhat hard to parse.

105: "When" should be in lowercase.

115: "and an experimental one".

126-129: Please consider rephrasing, it remains rather hard to follow.

185-186: I reckon it would be easier to follow if you used some sort of index to designate the three different sizes (e.g., "small", "medium", and "large"). The same designation could be carried later on when the comparisons are described.

210-211: I reckon it would read better as "something dogs did not need due to their height", or something to that effect.

233: As I mentioned last time, to my knowledge, numbers are always used as numerals when the unit is used as an abbreviation. So it can be "2 meters", "two meters", and "2 m", but not "two m".

235-237: I understand it is important to keep the subjects from getting satiated in order to increase the number of potential trials within a session, but would this not affect the animal's perception of the choice they made in the control conditions? Even assuming every individual would go for the highest amount of food under optimal conditions, if they were allowed to eat the same amount of food regardless of whether they choose the larger amount or not, then why would any of the choices be any better than the other? I reckon this deserves further comment, both in this section and the discussion.

262: It is unclear what sort of counterbalance is being discussed here.

273: "jump" > "skip"

298: Not sure if I understand how the "shape" variable was added to the model. The way I understood the sentence, the trial type would be a factor with 6 levels (one for each of the potential combination of stimuli), and the shape would be each of these stimuli. But this would be a problem when building the model matrix since, for instance, no data point could simultaneously have "circle vs. donut" in the "combination variable" and "dots" in the shape variable. So I suppose I did not quite get what you did here. Care to elaborate?

348: Not sure what is meant by the "(0.0001)"

458: Dogs in the wild live in fission-fusion societies, so I do not think the designation of "pack-living animal" is deserved. Either way, please provide references of such claims.

458-459: You may want to read this paper: https://doi.org/10.1007/s00265-017-2339-8 . It goes in the same direction to what you mention about "protecting one big piece of food from other conspecifics".

467: Is "Simona, Maria (43)" a citation? Not sure if this is clear in the text.

471-472: This sentence does not connect well with the rest of the paragraph, and I reckon it should be deleted.

474: A new paragraph should start in "seemingly".

496: I reckon "tricky" could be safely removed.

516: Which results in particular?

7. PLOS authors have the option to publish the peer review history of their article (what does this mean? ). If published, this will include your full peer review and any attached files.

**Do you want your identity to be public for this peer review?** For information about this choice, including consent withdrawal, please see our Privacy Policy .

Reviewer #1: No

Reviewer #2: **Yes: ** Dániel Rivas-Blanco

---

## [Author Response · Author response to Decision Letter 2]

16 May 2025

Reviewers' comments to the Author

Reviewer #1: Great job improving the clarity of the manuscript! Overall, the authors did a careful revision and adequately responded to my previous comments. The procedure is much clearer. However, I still have some comments and suggestions but much minor compared to the previous ones. As a general comment, the authors tend to use sentences and paragraphs that are too big and with too many ideas (this is even more salient in the discussion).

R: We have tried to divide the introduction and discussion into paragraphs so that each of them discusses only a single idea.

Abstract

p-2, line 31-32: be more specific than “some preference” or, at least, specify a preference for what.

R: We have added that the preference was for certain shapes, particularly the circle. See line 32.

p.2, line 34: at this point, the reader does not know what is the control condition. Instead of using the name of the condition, describe it as you did in the rest of the sentence.

R: We have added a description of the condition. See line 35.

Introduction

p. 4, line 68: processing numerical information involves a larger cognitive load compared to what? Also, what “this strategy” is referring to? You did not describe any strategy in the previous sentence.

R: We have added information and amended the sentence. See lines 69 and 70.

p.4, lines 73 to 76: Divide the sentence in key ideas. As it is, it is difficult to understand the message.

R: The sentence has been rewritten to be shorter and clearer. See lines 75-77 and 77-79.

p.4, line 82: why do you considered the use of number rather than of the continuous magnitudes a bias?

R: Thanks for noticing. We’ve reworded this sentence in line 85.

p.5, line 91: species rely on the ratio between what? the numerosity? the surface area? of the two stimuli. The information inside brackets is important to understand the results so it should be a regular sentence.

R: We have modified the sentence as suggested in lines 93-96.

p.5, line 97: decompose the sentence, there are too many ideas and information. How did the animals choose (response) and what was the correct response? Choosing the three item-set? End the sentence at this point. Then you can start the next sentence with results. This is only a suggestion:” Cats reliably learned to choose (...). But, when surface area was controlled, rats were indifferent between the two stimuli suggesting their previous preference was based on the surface are and not on quantity.”

R: Done, we have separated and clarified this section.

p.5, line 100: consider substituting “above 0.5 ratios” for “when the ratio between the two stimuli was greater than 0.5”. In the same sentence, I did not understood what the authors meant with “both numerical and continuous magnitude differences”.

R: We have substituted the sentence as suggested and we have added information for clarification. Other changes have been done as per the above comment, but this specific one can be found in lines 105-110.

p.5, line 105: Explain what you mean by “counterbalanced” in this case.

R: Counterbalanced was not the correct word replaced for the correct procedure description in lines 111-114.

p.6, line 114: The clarity of this paragraph can be increased.

R: We have summarized the paragraph to improve its clarity. See lines 115-129.

Methods

p.6, line 127 to 129: weight at least 10 kg instead of “over 10kg” and the information that comes next should not be part of the same paragraph. Suggestion: “This last constraint serve to….”.

R: Thanks for noticing this rather important detail. We have corrected the section as suggested in line 138.

p. 7, line 141: “details of”

R: Correct. We have amended the sentence which is in now line 152.

Table 1: I believe there is a better way to present the table. Since there is much more information regarding cats and dogs, I would suggest a table for those and the information regarding the humans can either be in a separate, simpler table or in the text. Below there is a table suggestion with only some information just to exemplify. You may need to switch the lines with columns due to the space occupied when more info is added.

Age Sex

M (SD) Range Female Male

Neutered Intact Neutered Intact

Cats 6.4 (3.7) 1 – 14 9 1 12 4

Dogs 4.1 (3.1) 1 – 13 11 4 9 4

R: Thank you for the suggestion, we have removed humans from the table and added their information to the text. The table is now presented in a similar manner to the reviewer’s example. See new Table 1 after line 155.

p. 10, line 189: the ratio between the surface areas of the two food portions was 0.5

R: We have changed the sentence as suggested. See line 204.

p.14, line 249: the experimenter asked the participant’s initials, age….

R: Done. Line 264.

p. 14, line 260: what is S1?; “completed trials of three different conditions”

R: We have modified the sentence. S1 is the supplementary information file which contains the instructions for the human participants and can be found at the end of the document, under Supporting information. Although the link to the actual file may not have been correctly provided by the submission system. In any case, we made sure the S1 is in the new revision.

p.14, line 263: “a total of 28 trials, x of each condition”

R: We have now added the number of trials for each condition. Lines 278-280.

p.15: Four trials of Control A plus 4 of Control B plus 6 of Experimental do not comprise 28 trials as stated before. I anticipate there were two trial-blocks?!

R: We have removed this as it was from a previous version of the MS and did not make sense anymore.

p.15, line 272: Suggestion: “If participants took more than 10 s to click on either option (i.e., the maximum choice latency allowed), the trial was cancelled and the next trial started immediately.

R: We have used the reviewer’s suggestion to modify some parts of our original sentence in lines 288-292.

Results

Instead of “latency of choice”, use “latency to choose”. Be careful when using “female dogs’ choice for the larger” or “human participants’ choice of the larger”, it adds complexity to the information.

R: We have changed “latency of choice” to “latency to choose” as suggested. We have also restructured the sentences mentioned to make them less complex (for instance, in line 340).

Discussion

In general, paragraphs are too long and with too many ideas. Try to divide them.

R: We have divided the paragraphs throughout the discussion to make them more concise.

p.23, line 426: substitute extremely by significantly

R: We have substituted the word in line 451.

p.23, line 428: According to the first part of the sentence, I was expecting the second part to be something like: while cats did not showed a preference for any particular shape or while cats were indifferent between shapes.

R: We have modified the sentence according to the reviewer’s expectation in line 453-454.

P23, line 430: “in the control condition” can be eliminated because you already said it in the beginning of the sentence.

R: We have removed this part. Line 456.

p.25, line 463: did you meant within each sample?.

R: The reviewer is right, we have changed it to within in line 501.

p.25, line 466: where the ratio between…

R: Done. Lines 505-506.

p.25, line 466: The difference in preference observed between male and female dogs…

R: Thank you for the suggestion, we have changed the sentence. Lines 506-507.

p.25, line 467: Did you check if there was any effect similar to that reported by Simona, Maria?

R: We added some text and a caveat regarding the small sample size after the contrast of our study to that of Simona, Maria’s. See lines 511-513.

p.25, line 473: dividing by sex; it might be possible; I think the conclusion of the sentence is odd. I would rather say that any differences may not be possible to observe in your sample because of its small size, mainly when dividing by sex.

R: We agree that the former sentence was odd. We have integrated the reviewer’s suggestion with the original idea to make it more understandable. See lines 516-520.

p.25, line 476: the discrimination is not between ratios is between two stimuli with a certain ratio between them. Be careful with this among the entire manuscript.

R: We have modified the sentence to make this point clear (line 522), and changed other instances along the manuscript where this may have been pointed out.

p.25, line 479: please explain what is the result suggests each of the two possibilities. What led you to those possibilities? It seems you are jumping a step in the rationale.

R: We have made more explicit the results that led us to this possibility but also have amended the text to focus only on the main one by removing lines 529-532.

p.26, line 490: …and their ratio change which….

R: Done. Line 540.

p.26, lines 496 and 497: three repetitions of the word making.

R: We have now changed two of these repetitions to make the sentence more clear. For instance, in line 547.

p.26, line 498: I believe words are missing from the sentence.

R: The reviewer is correct. We have added a few words to complete the sentence. See lines 548-549.

p.27, line 504: latencies

R: Done. Line 555.

p.27, line 506 and 507: when the control trials involved circles of different areas

R: We have changed the wording as suggested. Lines 558.

Reviewer #2: General comments

The main issue of this manuscript (and one that remains unchanged in this version) is that the authors do not exercise enough caution when comparing the different species. The paradigm used for both dogs and cats is similar enough to allow for a direct comparison, but humans get a completely different paradigm that involves no food, as well as instructions (that the animals, of course, do not get). Therefore, I think statistical analyses including data from all three species are inapropriate, and comparisons between humans and non-human animals should be kept to a minimum.

R: We agree with the reviewer. As suggested, we have opted for simpler statistical analyses for each individual species, avoiding direct comparisons between humans and non-human animals. Please see the new analysis in lines 313-332 and other changes throughout the manuscript.

The end of the introduction still feels rather bare, I think a paragraph in which the authors discuss the paradigm they used and how this connects to what the experiment is meant to test would be an important addition to the text.

R: As requested by R#1, we greatly revamped the next to last sentence to make it clearer. This now includes a statement of what the article is meant to evaluate. Please see lines 115-129.

Finally, the authors talk about ecological relevance being the driver behind the methods described. However, dogs are scavengers in nature, and they're well-suited for periods of scarcity. Under the abundance of food experienced by most house pets, optimal foraging (or, in this context, choosing the larger quantity) would arguably rarely be a priority. Particularly if, as the authors mention in the introduction, numerosity happens to be a particularly cognitively loading skill. These considerations do not need to be added to the manuscript, but I still thought they were worth discussing.

R: Given the concerns brought on by the reviewer, we opted to remove the statement about the food being an ecologically relevant stimulus and present it as can be read in lines 126-128.

In-line comments

103-107: Consider restructuring this sentence, it is somewhat hard to parse.

R: We have broken up the sentence into multiple ones and restructured it, we hope this improves the clarity.

105: "When" should be in lowercase.

R: We have corrected this mistake. Line 111.

115: "and an experimental one".

R: We have changed this as suggested. Line 125.

126-129: Please consider rephrasing, it remains rather hard to follow.

R: We have reworded the section so that it is less dense and can be followed easier. See lines 123-129.

185-186: I reckon it would be easier to follow if you used some sort of index to designate the three different sizes (e.g., "small", "medium", and "large"). The same designation could be carried later on when the comparisons are described.

R: We have incorporated this designation into the text so that the description of the methodology is easier to follow. See lines 198-200.

210-211: I reckon it would read better as "something dogs did not need due to their height", or something to that effect.

R: We appreciate the suggestion, we have used it to change the sentence. Lines 225-226.

233: As I mentioned last time, to my knowledge, numbers are always used as numerals when the unit is used as an abbreviation. So it can be "2 meters", "two meters", and "2 m", but not "two m".

R: The reviewer is right; we have corrected the sentence to “2 m”. See line 248.

235-237: I understand it is important to keep the subjects from getting satiated in order to increase the number of potential trials within a session, but would this not affect the animal's perception of the choice they made in the control conditions? Even assuming every individual would go for the highest amount of food under optimal conditions, if they were allowed to eat the same amount of food regardless of whether they choose the larger amount or not, then why would any of the choices be any better than the other? I reckon this deserves further comment, both in this section and the discussion.

R: We have added a limitation regarding the potential biases introduced by allowing the animals to consume the chosen stimuli. See lines 583-586.

262: It is unclear what sort of counterbalance is being discussed here.

R: We have described this in more detail to improve clarity.

273: "jump" > "skip"

R: Done. Line 292.

298: Not sure if I understand how the "shape" variable was added to the model. The way I understood the sentence, the trial type would be a factor with 6 levels (one for each of the potential combination of stimuli), and the shape would be each of these stimuli. But this would be a problem when building the model matrix since, for instance, no data point could simultaneously have "circle vs. donut" in the "combination variable" and "dots" in the shape variable. So I suppose I did not quite get what you did here. Care to elaborate?

R: The reviewer is correct; we have analyzed our model carefully and realized it poses problems due to some of its interactions. We have explored multiple alternative ideas and have decided to use a simpler form of analysis, binomial tests, for the results of the experimental trials; these give us equivalent results to the model while reducing complexity. This also removes the issue caused by comparing between the three species. See lines 313-332.

348: Not sure what is meant by the "(0.0001)"

R: This number referred to the proportion of errors in relation to the total number of trials. However, we realize it could be confused for a p-value and have thus removed it in line 370.

458: Dogs in the wild live in fission-fusion societies, so I do not think the designation of "pack-living animal" is deserved. Either way, please provide references of such claims.

R: While wild dog societies do have fission-fusion dynamics, we have seen that most studies investigating these species and their social structure describe their groups as “packs”. We have added references to this section in line 493.

458-459: You may want to read this paper: https://doi.org/10.1007/s00265-017-2339-8 . It goes in the same direction to what you mention about "protecting one big piece of food from other conspecifics".

R: We thank the reviewer for the suggestion we have read and incorporated the paper into the discussion as it offers evidence of food monopolization in dogs in lines 488-499.

467: Is "Simona, Maria (43)" a citation? Not sure if this is clear in the text.

R: Yes, this is a citation, but we had not formatted it properly. It has been fixed now, as can be seen in lines 508 and 513.

471-472: This sentence does not connect well with the rest of the paragraph, and I reckon it should be deleted.

R: We have rephrased this paragraph greatly, and in the processes, removed the sentence the reviewer

---

## [Decision Letter · Decision Letter 2]

20 Jul 2025

PONE-D-24-40472R2Continuous or discrete magnitudes? A comparative study between cats, dogs and humansPLOS ONE

Dear Dr. Rosetti,

Thank you for resubmitting your manuscript, and I sincerely apologize for the delay. Reviewer #1 is satisfied with your second round of revisions; however, Reviewer #2 has raised a few additional points, mostly grammatical but some related to the studies you referenced, that I would like you to consider. Once you have addressed Reviewer #2's comments, I will review your revisions and then deliver a final decision. I do not believe your final manuscript needs to go out for a third review, as many of the proposed changes can be made quickly and easily.

We look forward to receiving your revised manuscript.

Kind regards,

Brittany N. Florkiewicz, Ph.D.

Academic Editor

PLOS ONE

Journal Requirements:

Additional Editor Comments (if provided):

Thank you for resubmitting your manuscript, and I sincerely apologize for the delay. Reviewer #1 is satisfied with your second round of revisions; however, Reviewer #2 has raised a few additional points, mostly grammatical but some related to the studies you referenced, that I would like you to consider. Once you have addressed Reviewer #2's comments, I will review your revisions and then deliver a final decision. I do not believe your final manuscript needs to go out for a third review, as many of the proposed changes can be made quickly and easily.

Reviewers' comments:

Reviewer's Responses to Questions

**Comments to the Author**

1. If the authors have adequately addressed your comments raised in a previous round of review and you feel that this manuscript is now acceptable for publication, you may indicate that here to bypass the “Comments to the Author” section, enter your conflict of interest statement in the “Confidential to Editor” section, and submit your "Accept" recommendation.

Reviewer #1: All comments have been addressed

Reviewer #2: All comments have been addressed

2. Is the manuscript technically sound, and do the data support the conclusions?

Reviewer #1: Yes

Reviewer #2: Yes

3. Has the statistical analysis been performed appropriately and rigorously? 

Reviewer #1: Yes

Reviewer #2: Yes

4. Have the authors made all data underlying the findings in their manuscript fully available?

Reviewer #1: Yes

Reviewer #2: Yes

5. Is the manuscript presented in an intelligible fashion and written in standard English?

Reviewer #1: Yes

Reviewer #2: Yes

6. Review Comments to the Author

Reviewer #1: Most comments you will find in this review are minor and mostly related to wording.

Introduction

p.3, line 61: The definition of continuous magnitudes should be available earlier since it was already used a considerable number of times.

p. 4, line 94: consider specifying “…the ratio between the smaller and the larger stimuli”. If the ratio is between the larger over the smaller, the larger the ratio, the easier it is to discriminate.

p. 4, line 103: They could be based their choice on both cues and when one of them is changed (in this case, surface area) it is as if a completely new stimuli are presented and animals are indifferent.

p.5, line 112: Suggestion: “the use of numerosity with the use of continuous magnitudes of stimuli when discriminating between quantities and, the ones that do, entail methodological differences….”.

Methods

p. 8, line 158: Substitute “S” by “SD”.

p.9, line 185: Eliminate “one, four, or eight, …” from the sentence and add “ : ” following the parenthesis.

p. 9, line 191: Edit the sentence to make clear the use of the letters in line 192: “…three different diameters (d) … different surface areas (a)…”

p.13, line240 – 241: Confirm if you don’t need to add some commas.

p.13, line 246: “… and thus were given a 10-minute break.”

p. 13, line 254: Eliminate the word “study”.

p.14, line 272: “12 for the experimental condition.”

p.15, line 275: Suggestion: “In Control A, the surface areas were 29.22 cm2 and 14.61 cm2 thus, the ratio between the two stimuli was equal to 0.5.”

Analysis

p. 16, line 307: I think the authors meant to use “e.g.,” instead of “i.e.,”.

Results

p. 19, line 362: Edit the sentence: “… none of the shapes was chosen significantly more than the others, …”

p. 19, line 364: “… an effect on choice with subjects showing a significant preference for the circle over the crown…”

p. 19, line 367: “… revealing a significant preference for the circle …”

p. 20, Table 2: In the title, I believe the authors meant “Control and Experimental conditions” instead of trials.

Discussion

p. 23, line 426: I am not sure but I believe it is more accurate to use “faster” rather than “quicker”.

p. 24, lines 438 – 440: I was wondering if the fact that cats are generalist predators should not make them more sensitive to differences in shape and thus, I would expect the exact opposite result. The argument might be used both ways. To maximize their hunting they should be capable of clearly distinguishing between different preys.

p. 24, line 444: You don’t need the sentence starting with “We can…”. Start immediately with: “One possible explanation is that….”.

p.24, line 446: “a) they were well fed and b) they could…, food would be always available at the end of all trials”

p. 24, line 448: “Another possible explanation”

p. 24, p461: If the same phenomena was observed in cats, how can you use it as a possibility for a result found in dogs but not in cats?!

p.24, line 473: Was the difference between neutered and intact dogs significant? If not, it is not a difference. Moreover, I think you did not have a sample size big enough to say this.

p.24: In future studies, you can also use stimulus different from food with the animals.

Reviewer #2: I believe the authors have done a great job re-vamping the stats and addressing most of the issues that were found before. However, I still have a few minor comments remaining.

Line 69: "processing" is already used in this sentence, please consider a synonym.

Line 75: Please consider rephrasing the "They found that continuous [...]". I reckon "These studies found that continuous [...]" would be clearer and easier to follow.

Lines 112-114: This sentence should go before the new paragraphs in which the authors talk about previous studies with cats and dogs since, the way the sentence is formulated, it seems to imply that examples of such studies will be discussed straight after that. It would also serve to illustrate the different approaches that are used and why it would be necessary to use the same method for comparative purposes.

Line 114: On that topic, I am not sure if you want "they have relied on testing only a few species" to be a part of your message. Cats and dogs have been tested already. The one thing you provide over previous studies is a protocol that controls for shape of the stimulus and the comparative aspect of testing two species within the same protocol. Then, the question could also be why those two species in particular, etc.

Line 238: "Cueing" was correct. "Cueing" is "providing cues". "Queueing" is "standing in line".

LIne 272: "for experimental condition" → "for the experimental condition"

Lines 282-283: Consider rewording, not sure if I follow.

Lines 452-457: Please pay more attention to the literature: both "wild" and "free-ranging" dogs are dogs (and as far as I understand, they refer to the similar populations in this case), while dholes are not. Therefore, the reference about dholes seems wholly out of place, and so any wider references to "canids". Please consider reading further on studies done on free-ranging dogs.

Line 461: You do state that "similar phenomena was also observed in cats", yet in this study, cats did choose at random whenever presented with different shapes, which would go against the explanation you laid out in the lines above. Please elaborate on this.

Lines 461-463: Did the authors on those study also conclude that the largest item would facilitate monopolizing that resource? Please elaborate.

Line 469: You do continue with the same idea on the next paragraph. It may not be the best place to end the paragraph.

Lines 485-488: That being the case, would you not expect humans to show indifference for the different stimuli in much the same manner as cats did? Please elaborate.

7. PLOS authors have the option to publish the peer review history of their article (what does this mean? ). If published, this will include your full peer review and any attached files.

**Do you want your identity to be public for this peer review?** For information about this choice, including consent withdrawal, please see our Privacy Policy .

Reviewer #1: No

Reviewer #2: **Yes: ** Dániel Rivas-Blanco

---

## [Author Response · Author response to Decision Letter 3]

21 Aug 2025

Reviewer #1: Most comments you will find in this review are minor and mostly related to wording.

Introduction

p.3, line 61: The definition of continuous magnitudes should be available earlier since it was already used a considerable number of times.

R: The reviewer is correct. We have introduced a short definition in the first mention of continuous magnitudes, in line 47-48.

p. 4, line 94: consider specifying “…the ratio between the smaller and the larger stimuli”. If the ratio is between the larger over the smaller, the larger the ratio, the easier it is to discriminate.

R: We have replaced the first instance of ‘ratio’ with ‘difference’ as the sentence may be confusing. A larger ratio between the stimuli (100/99 = 0.99) is more difficult to discriminate than a smaller one (100/80 = 0.80).

p. 4, line 103: They could be based their choice on both cues and when one of them is changed (in this case, surface area) it is as if a completely new stimuli are presented and animals are indifferent.

R: We have added this argument into the description of the study, since we agree it is a possible explanation for the behavior shown.

p.5, line 112: Suggestion: “the use of numerosity with the use of continuous magnitudes of stimuli when discriminating between quantities and, the ones that do, entail methodological differences….”.

R: We have replaced the text with the reviewer’s suggestion.

Methods

p. 8, line 158: Substitute “S” by “SD”.

R: Done.

p.9, line 185: Eliminate “one, four, or eight, …” from the sentence and add “ : ” following the parenthesis.

R: Done.

p. 9, line 191: Edit the sentence to make clear the use of the letters in line 192: “…three different diameters (d) … different surface areas (a)…”

R: Done.

p.13, line240 – 241: Confirm if you don’t need to add some commas.

R: No commas, but we slightly rephrase the sentence.

p.13, line 246: “… and thus were given a 10-minute break.”

R: Done.

p. 13, line 254: Eliminate the word “study”.

R: Done.

p.14, line 272: “12 for the experimental condition.”

R: Done.

p.15, line 275: Suggestion: “In Control A, the surface areas were 29.22 cm2 and 14.61 cm2 thus, the ratio between the two stimuli was equal to 0.5.”

R: Done.

Analysis

p. 16, line 307: I think the authors meant to use “e.g.,” instead of “i.e.,”.

R: Done.

Results

p. 19, line 362: Edit the sentence: “… none of the shapes was chosen significantly more than the others, …”

R: Done.

p. 19, line 364: “… an effect on choice with subjects showing a significant preference for the circle over the crown…”

R: Done.

p. 19, line 367: “… revealing a significant preference for the circle …”

R: Done.

p. 20, Table 2: In the title, I believe the authors meant “Control and Experimental conditions” instead of trials.

R: Done.

Discussion

p. 23, line 426: I am not sure but I believe it is more accurate to use “faster” rather than “quicker”.

R: Done.

p. 24, lines 438 – 440: I was wondering if the fact that cats are generalist predators should not make them more sensitive to differences in shape and thus, I would expect the exact opposite result. The argument might be used both ways. To maximize their hunting they should be capable of clearly distinguishing between different preys.

R: In these sentences, our claim is not that they are not capable of distinguishing between shapes, but rather that cats do not do so in the present experiment. If every cat prefers a different shape, this may mask this ability. It is possible that each individual had its own preferred shape, but the number of experiments conducted under Experimental Conditions is insufficient to test this reliably. We know mention this more clearly in the text.

p. 24, line 444: You don’t need the sentence starting with “We can…”. Start immediately with: “One possible explanation is that….”.

R: Done.

p.24, line 446: “a) they were well fed and b) they could…, food would be always available at the end of all trials”

R: Done.

p. 24, line 448: “Another possible explanation”

R: Done.

p. 24, p461: If the same phenomena was observed in cats, how can you use it as a possibility for a result found in dogs but not in cats?!

R: We have reviewed the cited reference again and decided to remove its mention from the text since it is more focused on monopolization according to food quality, instead of size or shape like we tested in our protocol, and we understand it is confusing for the readers.

p.24, line 473: Was the difference between neutered and intact dogs significant? If not, it is not a difference. Moreover, I think you did not have a sample size big enough to say this.

R: We agree, the difference was not significant due to our limited sample size. Thus, we have decided to rewrite this sentence to clarify the findings.

p.24: In future studies, you can also use stimulus different from food with the animals.

R: We have added this to the limitations.

Reviewer #2: I believe the authors have done a great job re-vamping the stats and addressing most of the issues that were found before. However, I still have a few minor comments remaining.

Line 69: "processing" is already used in this sentence, please consider a synonym.

R: Done.

Line 75: Please consider rephrasing the "They found that continuous [...]". I reckon "These studies found that continuous [...]" would be clearer and easier to follow.

R: Done.

Lines 112-114: This sentence should go before the new paragraphs in which the authors talk about previous studies with cats and dogs since, the way the sentence is formulated, it seems to imply that examples of such studies will be discussed straight after that. It would also serve to illustrate the different approaches that are used and why it would be necessary to use the same method for comparative purposes.

R: We have moved this sentence further up to highlight the differences in previous studies, and we have also restructured the penultimate paragraph of our introduction so that the ideas are easier to follow.

Line 114: On that topic, I am not sure if you want "they have relied on testing only a few species" to be a part of your message. Cats and dogs have been tested already. The one thing you provide over previous studies is a protocol that controls for shape of the stimulus and the comparative aspect of testing two species within the same protocol. Then, the question could also be why those two species in particular, etc.

R: We have rewritten this section to emphasize the things mentioned by the reviewer (the comparison between species and the controlling of shape) instead of the focus on which species have been tested.

Line 238: "Cueing" was correct. "Cueing" is "providing cues". "Queueing" is "standing in line".

R: Done.

Line 272: "for experimental condition" → "for the experimental condition"

R: Done.

Lines 282-283: Consider rewording, not sure if I follow.

R: Done.

Lines 452-457: Please pay more attention to the literature: both "wild" and "free-ranging" dogs are dogs (and as far as I understand, they refer to the similar populations in this case), while dholes are not. Therefore, the reference about dholes seems wholly out of place, and so any wider references to "canids". Please consider reading further on studies done on free-ranging dogs.

R: We have removed references to other canids and focused solely on studies that discuss free-ranging dogs and their social behavior.

Line 461: You do state that "similar phenomena was also observed in cats", yet in this study, cats did choose at random whenever presented with different shapes, which would go against the explanation you laid out in the lines above. Please elaborate on this.

R: We have reviewed the cited reference again and decided to remove its mention from the text since it is more focused on monopolization according to food quality, instead of size or shape like we tested in our protocol, and we understand it is confusing for the readers.

Lines 461-463: Did the authors on those study also conclude that the largest item would facilitate monopolizing that resource? Please elaborate.

R: The authors describe another test in which a much smaller amount of food but with more pieces does not induce monopolization. However, it is important to consider this test was conducted only in dyads and not in a group setting. We try to address this and the need for more studies on the topic in lines...

Line 469: You do continue with the same idea on the next paragraph. It may not be the best place to end the paragraph.

R: Done.

Lines 485-488: That being the case, would you not expect humans to show indifference for the different stimuli in much the same manner as cats did? Please elaborate.

R: We believe our use of “idiosyncratically guided” was an oversimplification of the processes at hand and we have modified the sentence to clarify why we think the human participants had such preferences.

---

## [Editor Report · Decision Letter 3]

24 Aug 2025

Continuous or discrete magnitudes? A comparative study between cats, dogs and humans

PONE-D-24-40472R3

Dear Dr. Rosetti,

We’re pleased to inform you that your manuscript has been judged scientifically suitable for publication and will be formally accepted for publication once it meets all outstanding technical requirements.

Kind regards,

Brittany N. Florkiewicz, Ph.D.

Academic Editor

PLOS ONE

Additional Editor Comments (optional):

Thank you for addressing the minor revisions suggested by reviewer #2. The authors have effectively responded to all comments, and I am pleased to recommend this manuscript for publication. Congratulations!
---

## [Editor Report · Acceptance letter]

PONE-D-24-40472R3

PLOS ONE

Dear Dr. Rosetti,

I'm pleased to inform you that your manuscript has been deemed suitable for publication in PLOS ONE. Congratulations! Your manuscript is now being handed over to our production team.

Kind regards,

on behalf of

Dr. Brittany N. Florkiewicz

Academic Editor

PLOS ONE